# Svep1 is a binding ligand of Tie1 and affects specific aspects of facial lymphatic development in a Vegfc-independent manner

**Melina Hußmann[1], Dörte Schulte[1], Sarah Weischer[2], Claudia Carlantoni[3], Hiroyuki Nakajima[4], Naoki Mochizuki[4], Didier YR Stainier[3], Thomas Zobel[5], Manuel Koch[6], Stefan Schulte-Merker[1]***

[1]Institute of Cardiovascular Organogenesis and Regeneration, Faculty of Medicine, WWU Münster, Münster, Germany; [2]Münster Imaging Network, Cells in Motion Interfaculty Centre, Faculty of Biology, WWU Münster, Münster, Germany; [3]Max Planck Institute for Heart and Lung Research, Department of Developmental Genetics, Bad Nauheim, Germany; [4]Department of Cell Biology, National Cerebral and Cardiovascular Center Research Institute, Osaka, Japan; [5]Münster Imaging Network, Cells in Motion Interfaculty Centre, WWU Münster, Münster, Germany; [6]Institute for Dental Research and Oral Musculoskeletal Biology, Center for Biochemistry, Faculty of Medicine and University Hospital Cologne, University of Cologne, Cologne, Germany

**Abstract** Multiple factors are required to form functional lymphatic vessels. Here, we uncover an essential role for the secreted protein Svep1 and the transmembrane receptor Tie1 during the development of subpopulations of the zebrafish facial lymphatic network. This specific aspect of the facial network forms independently of Vascular endothelial growth factor C (Vegfc) signalling, which otherwise is the most prominent signalling axis in all other lymphatic beds. Additionally, we find that multiple specific and newly uncovered phenotypic hallmarks of *svep1* mutants are also present in *tie1*, but not in *tie2* or *vegfc* mutants. These phenotypes are observed in the lymphatic vasculature of both head and trunk, as well as in the development of the dorsal longitudinal anastomotic vessel under reduced flow conditions. Therefore, our study demonstrates an important function for Tie1 signalling during lymphangiogenesis as well as blood vessel development in zebrafish. Furthermore, we show genetic interaction between *svep1* and *tie1* in vivo, during early steps of lymphangiogenesis, and demonstrate that zebrafish as well as human Svep1/SVEP1 protein bind to the respective Tie1/TIE1 receptors in vitro. Since compound heterozygous mutations for *SVEP1* and *TIE2* have recently been reported in human glaucoma patients, our data have clinical relevance in demonstrating a role for SVEP1 in TIE signalling in an in vivo setting.

*For correspondence:
schultes@ukmuenster.de

## Editor's evaluation

This study presents strong and compelling evidence that the extra-cellular matrix protein SVEP-1 interacts with the TIE1 receptor to promote aspects of lymphangiogenesis that are independent of canonical VEGF-C signaling. Using zebrafish models to show genetic interactions and cells to provide evidence of biochemical interaction, the study shows a functional requirement for these genes/proteins in specific aspects of lymphangiogenesis. These novel findings will be of interest to developmental and cell biologists and to those studying lymphatic disease as it potentially provides novel therapeutic targets.

## Introduction

The lymphatic system is part of the vasculature and provides essential functions for tissue fluid homeostasis, absorption of dietary fats, and immune surveillance. Malfunction of the lymphatic vasculature can lead to severe lymphedema, obesity, or chronic inflammatory diseases (*Mäkinen et al., 2021*; *Oliver et al., 2020*). Since treatment options are rare and often only transiently effective, understanding the molecular mechanisms driving lymphangiogenesis is a prerequisite for the development of new therapeutic approaches (*Mäkinen et al., 2021*). To that end, mice and zebrafish have served as popular model organisms to study the development of lymphatic vessels and are commonly used for analyzing the underlying genetic and molecular mechanisms (*Mäkinen et al., 2007*; *Padberg et al., 2017*; *van Impel and Schulte-Merker, 2014*). Furthermore, many genes that are essential for lymphangiogenesis in zebrafish are evolutionarily conserved. Their inactivation leads to lymphatic defects in zebrafish and mice, and mutations in their orthologues are causative for human diseases (*Alders et al., 2009*; *Gordon et al., 2013*; *Hogan et al., 2009*; *Mauri et al., 2018*; *Wang et al., 2020*). In the trunk vasculature of the zebrafish, so-called lympho-venous sprouts arise from the posterior cardinal vein at 32 hours post-fertilization (hpf). They migrate dorsally and either remodel an intersegmental artery into a vein, or they migrate along the so-called horizontal myoseptum (HM) as parachordal lymphangioblasts (PLs) at 2 days post fertilization (dpf). At 3 dpf, PLs migrate dorsally and ventrally to form the trunk lymphatic vasculature, consisting of the dorsal longitudinal lymphatic vessel, the intersegmental lymphatic vessels, and the thoracic duct (TD) (*Hogan et al., 2009*; *Hogan and Schulte-Merker, 2017*; *Padberg et al., 2017*). A separate lymphatic network, the facial lymphatics, arises in a distinctly different manner, originating from three progenitor populations: (1) the primary head sinus-lymphatic progenitors (PHS-LP), (2) a migratory angioblast cell near the ventral aorta, and (3) the major population sprouting from the common cardinal vein (CCV) (*Eng et al., 2019*). These progenitor populations proliferate, migrate and connect to each other in a relay-like mechanism (*Eng et al., 2019*). A third lymphatic bed is composed of the brain lymphatic endothelial cells (BLECs), which are single endothelial cells residing within the leptomeningeal layer of the zebrafish brain and that arise from the choroidal vascular plexus (*Bower et al., 2017*; *van Lessen et al., 2017*; *Galanternik et al., 2017*). During larval stages, BLECs are often positioned next to meningeal blood vessels and stay at the distal periphery of the optic tectum and other brain regions (*van Lessen et al., 2017*). However, molecular mechanisms supporting the development of BLECs and facial lymphatics still need to be examined in more detail.

The best-studied pathway driving lymphangiogenesis comprises the growth factor Vascular Endothelial Growth Factor C (VEGFC), which is secreted as a pro-form that is processed through the concerted activity of Collagen and Calcium-Binding EGF domain-containing protein 1 (CCBE1) (*Bos et al., 2011*; *Hogan et al., 2009*; *Jeltsch et al., 2014*; *Le Guen et al., 2014*; *Roukens et al., 2015*) and a disintegrin and metalloproteinase with thrombospondin motifs (Adamts) 3/14 (*Jeltsch et al., 2014*; *Wang et al., 2020*) in the extracellular space. Fully processed VEGFC binds to its receptor VEGFR3 as well as VEGFR2 and induces lymphangiogenesis (*Joukov et al., 1997*; *Karkkainen et al., 2004*). Apart from the VEGFC/VEGFR3 pathway, TIE-ANG signalling was shown to be essential for lymphangiogenesis and vessel remodelling in mice and humans. This signalling cascade is composed of two receptor tyrosine kinases, tyrosine-protein kinase receptor 1 (TIE1) (*Partanen et al., 1992*) and tyrosine endothelial kinase (TEK), also known as tyrosine-protein kinase receptor 2 (TIE2) (*Dumont et al., 1993*), and multiple angiopoietin ligands including angiopoietin 1 (ANG 1) (*Davis et al., 1996*; *Suri et al., 1996*) and angiopoietin 2 (ANG 2) (*Maisonpierre et al., 1997*). In mammals, TIE signalling is activated through binding of Angiopoietins to TIE2 (*Davis et al., 1996*; *Maisonpierre et al., 1997*). TIE1 can either block or activate the signalling cascade in a context-dependent manner by forming heterodimers with TIE2 (*Hansen et al., 2010*; *Marron et al., 2000*; *Saharinen et al., 2005*; *Savant et al., 2015*; *Seegar et al., 2010*). *Tie1* knockout mice display haemorrhages from E13.5 to P0, which lead to death and are preceded by lymphatic defects and edema formation from E12.5 onwards (*D'Amico et al., 2010*). In contrast to *Tie1* mutant mice, *Tie2* mutant mice die already at E9.5–10.5 due to defective cardiac development and vascular remodelling (*Dumont et al., 1994*; *Sato et al., 1995*). Conditional knockout of *Tie2* in lymphatic cells revealed the importance of TIE2 for lymphatic vessel development in mice especially for Schlemm's canal formation (*Kim et al., 2017*; *Thomson et al., 2014*). Recently, Korhonen et al. showed that conditional *Tie1* deletion, *Tie1;Tie2* double deletion and *Ang2* blocking resulted in impaired postnatal lymphatic capillary network development in

mice (*Korhonen et al., 2022*). In zebrafish, *tie2* mutants do not have any overt vascular defects (*Gjini et al., 2011*; *Jiang et al., 2020*), while *tie1* mutants show cardiac morphogenesis and vascular defects (*Carlantoni et al., 2021*).

In 2017, a new key player in lymphangiogenesis was discovered through genetic screens in zebrafish: *sushi, von Willebrand factor type A, EGF, and pentraxin domain-containing protein 1* (*svep1*), also referred to as *polydom* (*Karpanen et al., 2017*; *Morooka et al., 2017*). *Svep1* encodes a large extracellular matrix molecule, with a total of 3571 amino acids and a variety of protein domains. The C terminal half of Svep1 mainly consists of complement control protein (CCP), also called sushi domain, repeats and EGF domains, indicating a possible role in protein-binding stabilization. *Svep1−/−* mice show normal development of the primitive lymphatic plexus until E12.5, but then fail to undergo remodelling of lymphatic vessels and formation of lymphatic valves at later embryonic stages, accompanied by edema formation and death postnatally (*Karpanen et al., 2017*; *Morooka et al., 2017*). Recently, Michelini et al. reported possible implications of *SVEP1* in lymphedema formation in human patients, underlining the importance of SVEP1 for the lymphatic vasculature (*Michelini et al., 2021*). Additionally, SVEP1 is also required for Schlemm's canal formation in mice (*Thomson et al., 2021*). In zebrafish, *svep1* mutants exhibit a near-complete loss of the TD, demonstrating an essential function during lymphangiogenesis in zebrafish (*Karpanen et al., 2017*; *Morooka et al., 2017*).

In the present study, we show defects in the lymphatic head vasculature in *svep1* mutants, comprising a variable loss of BLECs and a specific facial lymphatic phenotype, which is complementary to the phenotypes observed in mutants of Vegfc/Vegfr3 pathway members. Therefore, we identified a lymphatic structure in the zebrafish that, in contrast to all other lymphatic structures, forms independently of the Vegfc/Vegfr3 pathway, but depends on Svep1.

Murine SVEP1 has been shown to bind to the α9 form of integrin (ITGA9) as well as the TIE2 ligands ANG1 and ANG2 in vitro (*Morooka et al., 2017*; *Sato-Nishiuchi et al., 2012*). However, until now, putative interaction partners of Svep1 have not been confirmed in vivo. In the present study, we first characterized novel lymphatic and blood vasculature defects of *tie1* mutants, and subsequently realized that all phenotypic traits are shared between *tie1* and *svep1* mutants. These observations raised the question whether Svep1 and Tie1 interact, a notion that we tested both genetically and on a protein biochemistry level. Our results provide the first in vivo evidence for *svep1* and *tie1* genetic interaction, thus placing Svep1 as an important regulator of Tie1 function. Additionally, we show the interaction of SVEP1 and TIE1 in vitro for the respective versions of the zebrafish and human proteins. Since recent clinical data suggested SVEP1 as a genetic modifier of TIE2-related primary congenital glaucoma (PCG) (*Young et al., 2020*), our results have clinical relevance and will further help to understand the molecular basis of PCG.

## Results

### Svep1 is required for facial collecting lymphatic vessel formation in a Vegfc-independent manner

Since *svep1* mutants had previously been analyzed for lymphatic defects only in the trunk vasculature, we examined the head vasculature of *svep1* mutants to detect further possible malformations of the lymphatic system. At 5 dpf we observed that *svep1* mutants showed specific facial lymphatic defects, which seemed to be complementary to the facial lymphatic defects found in mutants of the Vegfc/Vegfr3 pathway members (*Figure 1A*). While mutants for Vegfc/Vegfr3 pathway members like *ccbe1*, *adamts3/14*, and *vegfc* retained the facial collecting lymphatic vessel (FCLV) (red dotted line in *Figure 1A, B*) but lacked all other structures of the facial lymphatics, *svep1* mutants showed a specific loss of the FCLV. All other parts of the mature facial lymphatic network (including lymphatic branchial arches, lateral facial lymphatic, medial facial lymphatic, and otolithic lymphatic vessel (blue dotted line in *Figure 1A*)) were only partially reduced in *svep1* mutants. Although the formation of the FCLV was strongly affected in all *svep1* mutants analyzed, the severity of the defects of facial lymphatic structures varied between individual *svep1* mutant embryos (*Figure 1—figure supplement 1*). Only simultaneous interference of both the Vegfc and Svep1 signalling pathways completely blocked the development of all facial lymphatic structures (*Figure 1—figure supplement 2*). To further characterize the differential roles of Svep1 and Vegfc during the formation of the facial lymphatic network, we examined the expression patterns of *svep1* and *vegfc* during sprouting of the PHS-LP, the progenitor

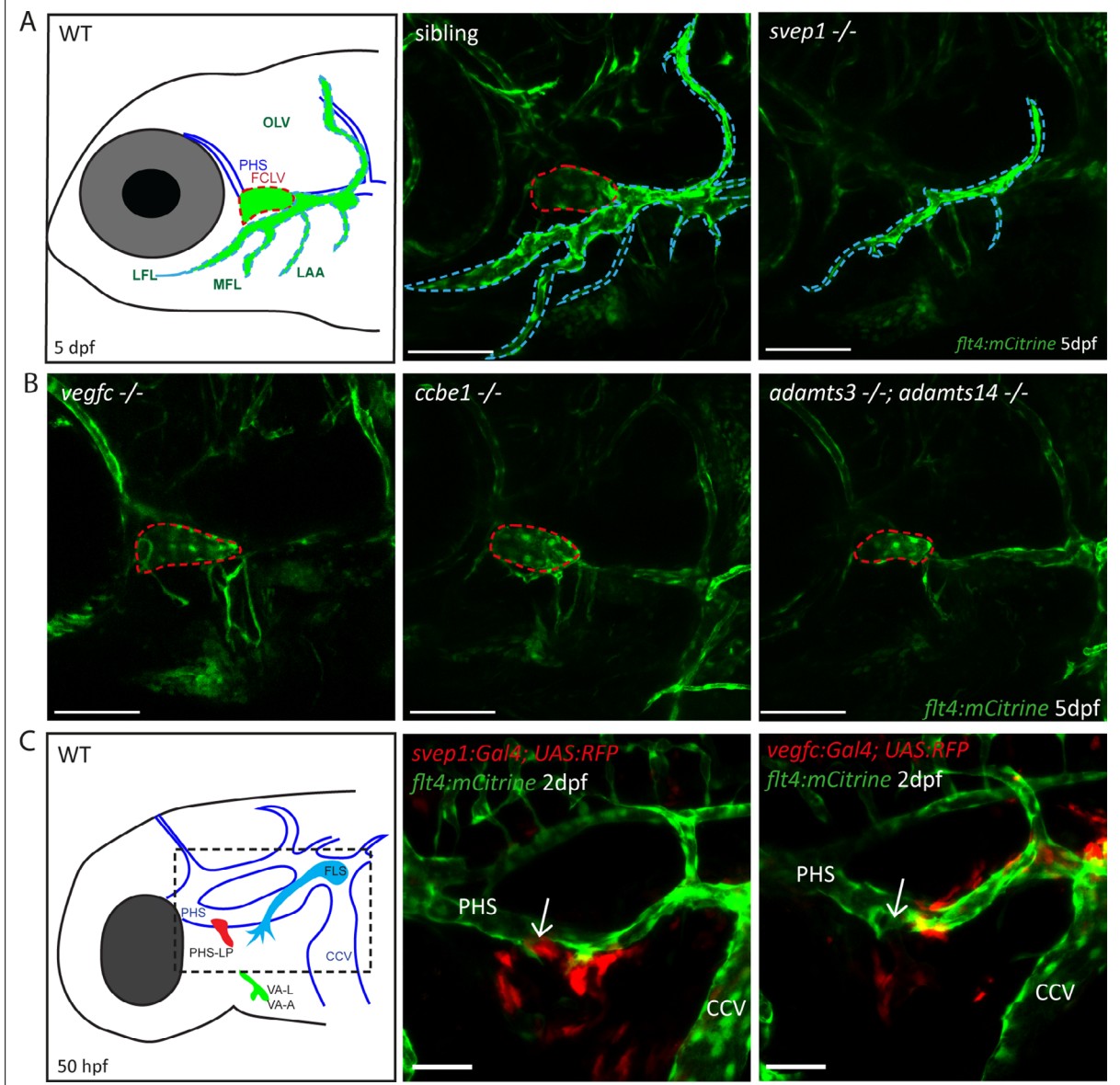

**Figure 1.** Svep1 is required for the development of the FCLV, in a Vegfc-independent manner. (**A**) Schematic representation of facial lymphatic network at 5 dpf and maximum intensity projection of confocal images of *flt4:mCitrine* positive *svep1* mutants (*n* = 10) and siblings (*n* = 6), highlighting facial lymphatic structures at 5 dpf. Scale bar = 100 μm. Note the absence of the FCLV (red dotted line) in *svep1* mutants whereas other facial lymphatic structures are less strongly affected (OLV, LFL, MFL, and LAA marked by blue dotted lines). (**B**) Confocal images of *flt4:mCitrine* positive facial lymphatics in *vegfc* (*n* = 19), *ccbe1* (*n* = 5), and *adamts3;adamts14* (*n* = 2) mutants at 5 dpf. Scale bar = 100 μm. (**C**) Confocal images of *svep1* and *vegfc* expression domains during sprouting from the PHS at 2 dpf, with schematic representation of different lymphatic progenitor populations. *svep1* is expressed in close proximity to sprouting PHS-LPs, while *vegfc* expressing cells are more concentrated on the LECs arising from the CCV. Arrows point to sprouting PHS-LP. Scale bar = 50 μm. Expression patterns were confirmed in six embryos each (*Figure 1—figure supplement 3*). CCV, common cardinal vein; dpf, days post-fertilization; FCLV, facial collecting lymphatic vessel; FLS, facial lymphatic sprout; hpf, hours post-fertilization; LAA, lymphatic branchial arches; LEC, lymphatic endothelial cell; LFL, lateral facial lymphatic; MFL, medial facial lymphatic; OLV, otolithic lymphatic vessel; PHS, primary head sinus; PHS-LP, primary head sinus lymphatic progenitor; VA, ventral aorta; VA-A, ventral aorta angioblast; VA-L, ventral aorta lymphangioblast; WT, wildtype.

The online version of this article includes the following figure supplement(s) for figure 1:

**Figure supplement 1.** Facial lymphatic phenotype of *svep1* mutant embryos.

**Figure supplement 2.** Combined loss of *svep1* and *ccbe1* leads to a loss of all facial lymphatic structures.

**Figure supplement 3.** *svep1* and *vegfc* expression.

cells of the FCLV, at 50 hpf using transgenic reporter lines. We detected *svep1* expression in cells juxtaposed to the sprouting LECs around the PHS, which later will form the FCLV, while *vegfc* expression was more restricted to the lateral facial lymphatic sprout arising from the CCV in all embryos analyzed (*Figure 1C*, *Figure 1—figure supplement 3*). Taken together, these observations indicate a Vegfc-independent role of Svep1 during the development of distinct aspects of the facial lymphatics.

## Svep1 is essential for sprouting of BLECs and is expressed in close proximity to BLECs

Since Svep1 is required for the formation of facial lymphatic structures (*Figure 1*), we wondered whether it is also involved in the development of an additional set of lymphatic endothelial cells, the BLECs. In mutants of the Vegfc/Vegfr3 pathway, BLECs are completely absent (*Bower et al., 2017*;

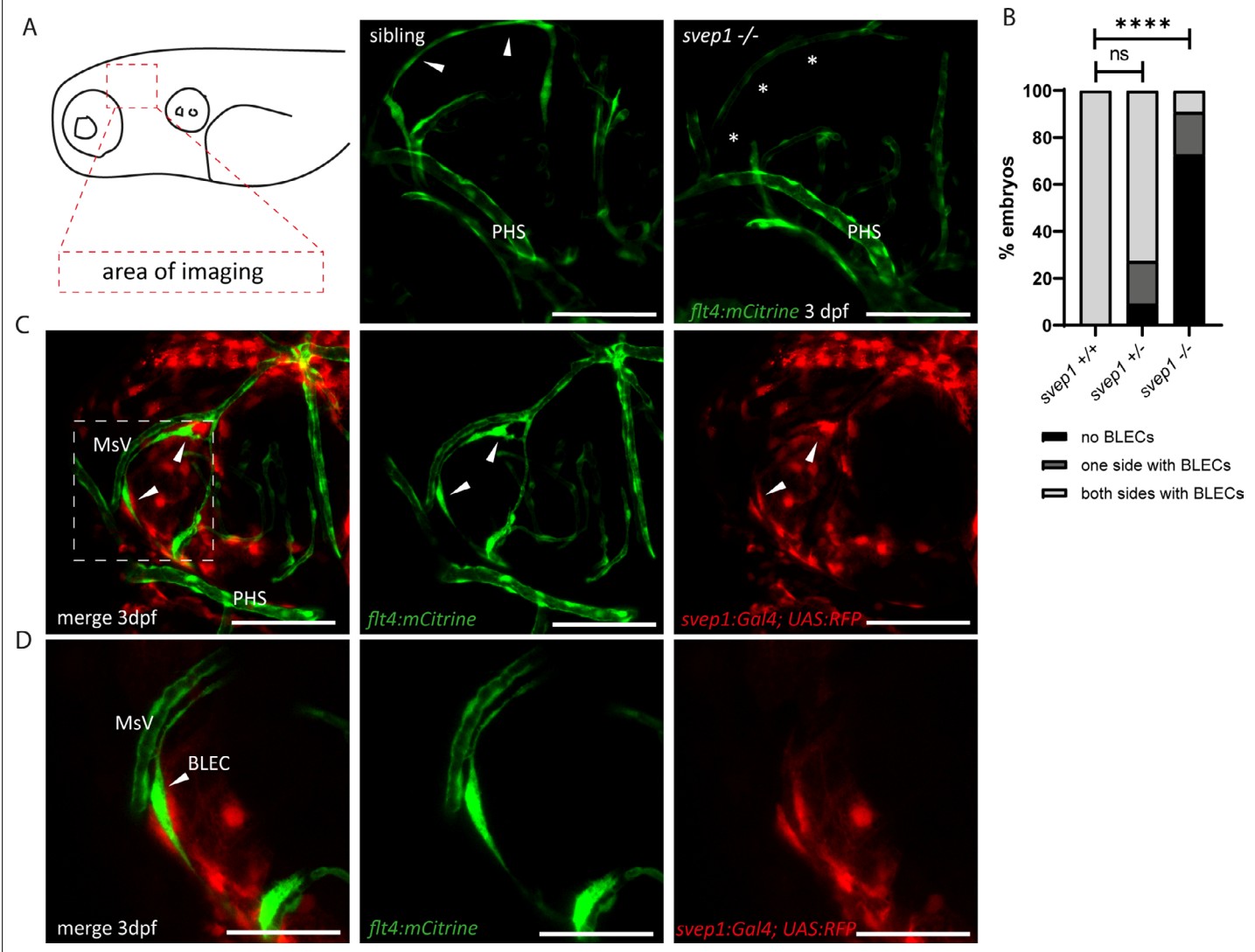

**Figure 2.** Svep1 is required for the sprouting of BLECs. (**A**) Confocal images of sprouting BLECs, marked by *flt4:mCitrine*, at 3 dpf in *svep1* mutants and siblings. Asterisks mark missing BLECs in *svep1* mutants. Scale bar = 100 μm. (**B**) Quantification of BLECs at 3 dpf on each side of the embryo showed that *svep1* mutants have significantly less BLECs on one or both sides of the brain hemispheres compared to siblings. For statistical analysis, no BLECs were counted as 0, BLECs being present on only one hemisphere as 1, whereas BLECs being detectable on both brain hemispheres were included as 2, for each embryo (*svep1+/+*: n = 10; *svep1+/−*: n = 12; *svep1−/−*: n = 12). Mann–Whitney test was applied for statistical analysis. Values are presented as means ± standard deviation (SD), ****p < 0.0001, ns = not significant. Scale bar = 100 μm. (**C**) Confocal images of *svep1:Gal4; UAS:RFP*, showing *svep1* expression immediately adjacent to BLECs, marked by arrowheads, at 3 dpf. Scale bar = 100 μm. (**D**) Magnification and reduced stack numbers of boxed area in (**C**). Arrowhead marks BLEC. Scale bar = 50 μm. BLEC, brain lymphatic endothelial cell; dpf, days post-fertilization; MsV, mesencephalic vein; PHS, primary head sinus;.

*van Lessen et al., 2017*). In *svep1* mutants, BLECs were found to be absent in most cases, but some embryos showed either reduced numbers or – in rare cases – even wildtype-like numbers of BLECs at 3 dpf (*Figure 2A, B*). In line with the idea that *svep1* is required for the sprouting and migration of BLECs, we observed *svep1* expressing cells in close proximity to the migrating BLECs at 3 dpf (*Figure 2C, D*). Thus, there is close juxtaposition of *svep1* expressing cells with migrating LECs in all developing lymphatic structures examined, including the PLs in the trunk (*Karpanen et al., 2017*).

### *svep1* and *tie1* mutants show near-identical lymphatic defects

Murine SVEP1 has been shown to bind the TIE2 ligands ANG1 and ANG2 in vitro and to regulate expression of *Tie1* as well as *Tie2* (*Morooka et al., 2017*). It also has been suggested to play a role in *TIE2*-related PCG (*Young et al., 2020*). Hence, we wanted to investigate the role of Tie signalling in zebrafish lymphangiogenesis in order to assess potential interactions with *svep1* in an in vivo situation. Lymphatic defects have not been previously reported in zebrafish mutants for either *tie1* or *tie2* (*Carlantoni et al., 2021*; *Gjini et al., 2011*; *Jiang et al., 2020*). Given the fact that there seems to be a very specific requirement for *svep1* in FCLV development, we analysed facial lymphatic structures of *tie1* and *tie2* mutants in direct comparison to *svep1* mutants. Since *tie1* mutants developed strong edema at 4 dpf (data not shown), we focused our analysis on lymphatic phenotypes at 2 and 3 dpf to exclude secondary effects on the lymphatic vasculature. Significantly, *tie1* mutant embryos showed the same facial lymphatic defects as *svep1* mutant embryos at 3 dpf (*Figure 3A*), with the FCLV being strongly affected. We confirmed this observation also in a *lyve1:DsRed* transgenic background (*Figure 3—figure supplement 1*). This finding suggests that Tie1, either independently or in concert with Svep1 is responsible for FCLV formation in a Vegfc-independent manner. Examining other lymphatic cells, we found that *tie1* mutants did not show any BLECs at 3 dpf and exhibited significantly reduced numbers of PLs at 2 dpf, similar to *svep1* mutants (*Figure 3B–E*). Importantly, *tie2* mutant embryos, when examined for the same anatomical features, were found to display normal facial lymphatics, BLECs and PL numbers (*Figure 3A–C and E*). Taken together, these findings demonstrate that loss of *tie1*, but not *tie2*, results in lymphatic defects highly similar to the ones seen in *svep1* mutants, indicating that Svep1 constitutes an essential component acting in the Tie1 pathway.

### *tie1* and *svep1* mutants display identical PL cell migration and survival defects

PLs first migrate along the HM and then start to migrate dorsally and ventrally along arteries to form the DLLV or the TD, respectively. Previously, it was shown that PLs in *svep1* mutants fail to migrate dorsally or ventrally and rather remain at the HM (*Karpanen et al., 2017*). Here, we compared PL migration in *svep1* and *tie1* mutants using overnight imaging from 2.5 to 3.5 dpf to analyse if PLs in *tie1* mutants phenocopy the PL migration defects of *svep1* mutants (*Figure 4A–L*, *Figure 4—videos 1–3*). While around 40–50% of PLs in sibling embryos migrated along the artery, only 11% of PLs in *tie1* and *svep1* mutants showed migration in either dorsal or ventral direction along the artery (*Figure 4M, N*). Additionally, we observed around 33% apoptotic PLs in *tie1* mutants and 55% in *svep1* mutants. These apoptotic events could be a consequence of failed migration, or could be due to decreased survival as a direct consequence of absent Svep1 or Tie1 activity. To further characterize migration of PLs in *svep1* and *tie1* mutants, we tracked and plotted the migration route of individual PLs (*Figure 4O, Q*, *Figure 4—figure supplements 1 and 2*) and quantified the migration distance in the Y direction (i.e. migration in dorsal or ventral direction), mean velocity and total migration distance in *tie1* and *svep1* mutants (*Figure 4P, R*). PLs in *svep1* as well as in *tie1* mutants showed significantly less migration in ventral and dorsal directions compared to siblings, while the mean velocity and total migration distance were unchanged. Therefore, we can conclude that Svep1 and Tie1 are required for PL migration along the arteries in dorsal or ventral direction. Since we could observe the same specific migratory defects in both *svep1* and *tie1* mutants, these results further support a possible cross-talk between both proteins.

### *tie1* mutants show blood vascular defects under reduced flow conditions

While *svep1* mutants were initially identified on the basis of their lymphatic phenotype (*Karpanen et al., 2017*), Coxam et al. recently showed that *svep1* mutant embryos display unique vascular

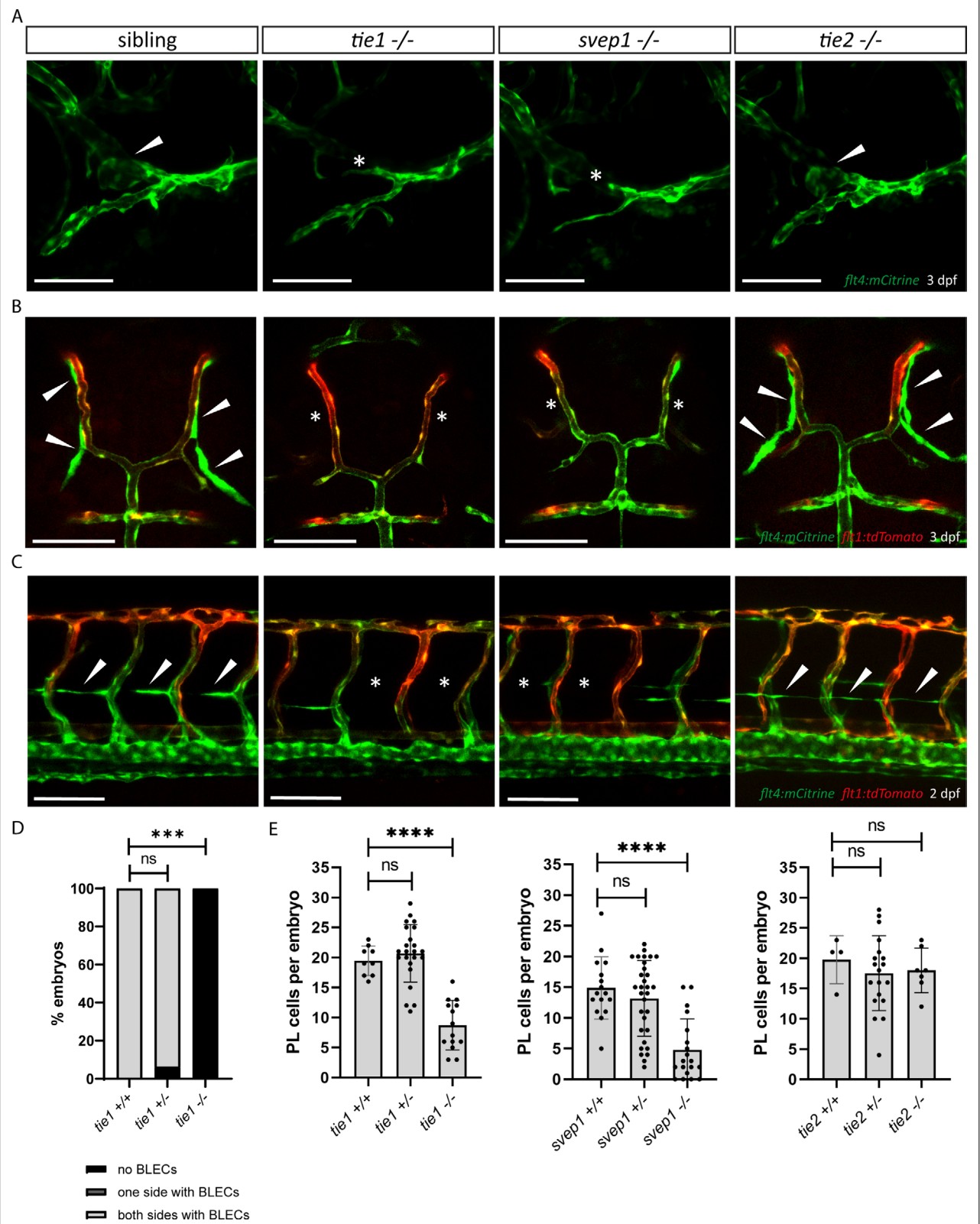

**Figure 3.** *tie1* mutants phenocopy the loss of *svep1*, while *tie2* is dispensable for lymphangiogenesis. (**A**) Facial lymphatics at 3 dpf in *flt4:mCitrine* positive *tie1*, *svep1* and *tie2* mutants and sibling embryos (lateral view). Arrowheads point to FCLV and asterisks indicate the absence of FCLV. Scale bar = 100 µm. (**B**) *flt4:mCitrine; flt1:tdTomato* positive dorsal head vasculature in *tie1*, *svep1*, and *tie2* mutants and in siblings at 3 dpf (dorsal view). In *svep1* and *tie1* mutants (but not in *tie2* mutants) the presence of BLECs is strongly reduced. Arrowheads point to BLECs and asterisks indicate areas lacking

*Figure 3 continued on next page*

Figure 3 continued

BLECs. Scale bar = 100 µm. (**C**) Confocal images of PL cells, indicated by arrowheads, at 2 dpf in *flt4:mCitrine; flt1:tdTomato* positive *tie1*, *svep1*, and *tie2* mutants and siblings, showing reduced PL numbers in *svep1* and *tie1* mutants. Asterisks indicate missing PLs. Scale bar = 100 µm. (**D**) Quantification of the presence of BLECs in *tie1* mutants compared to siblings. (*tie1+/+*: n = 6; *tie1+/−*: n = 16; *tie1−/−*: n = 10) Mann–Whitney test was applied for statistical analysis. ***p = 0.001, ns = not significant. (**E**) Quantification of PL cell numbers in *tie1* (*tie1+/+*: n = 9; *tie1+/−*: n = 23; *tie1−/−*: n = 14), *svep1* (*svep1+/+*: n = 16; *svep1+/−*: n = 31; *svep1−/−*: n = 19), and *tie2* (*tie2+/+*: n = 17; *tie2+/−*: n = 27; *tie2−/−*: n = 16) mutants compared to siblings. Mann–Whitney test was applied for statistical analysis. Values are presented as means ± standard deviation (SD), ****p < 0.0001, ns = not significant; BLEC, brain lymphatic endothelial cell; dpf, days post-fertilization; FCLV, facial collecting lymphatic vessel; PL, parachordal lymphangioblast.

The online version of this article includes the following figure supplement(s) for figure 3:

**Figure supplement 1.** Facials lymphatics of *svep1* and *tie1* mutant embryos.

defects under reduced flow conditions (*Coxam et al., 2022*). Treatment of embryos with 0.014% tricaine between 30 and 48 hpf leads to incomplete formation of the dorsal longitudinal anastomotic vessel (DLAV) with gaps and non-lumenized DLAV segments at 2 dpf in *svep1* mutant embryos. This phenotype is accompanied by increased Vegfa/Vegfr signalling and increased number of Apelin positive tip cells (*Coxam et al., 2022*). To investigate if *tie1* mutants mimic this very specific and unusual vascular defect, we treated embryos from *tie1* heterozygous parents with 0.014% tricaine between 30 and 48 hpf, and subsequently imaged the intersegmental vessels in the trunk. Our analysis showed that *tie1* mutants treated with tricaine exhibited significantly more gaps and fewer lumenized DLAV segments (*Figure 5D*) compared to both untreated *tie1* mutants (*Figure 5B*) and treated siblings (*Figure 5C, E, F*), suggesting that Svep1 and Tie1 might interact not only in lymphangiogensis but also during blood vessel development. For *tie2* and *vegfc* mutants we did not observe any defects in DLAV formation upon tricaine treatment, indicating that this phenotype is specific for loss of Svep1 and Tie1 (*Figure 5—figure supplement 1*). Additionally, upon tricaine treatment, and even in untreated conditions, *apelin* expressing ECs were increased in ISVs of *tie1* mutants as already shown for *svep1* morphants treated with tricaine in *Coxam et al., 2022* (*Figure 5G–J*). Since we observed increased *apelin* expressing ECs in *tie1* mutants already in untreated conditions, we investigated if *svep1* morphants also show increased *apelin* expression even without tricaine treatment (*Figure 5I, J*). *svep1* morphants already showed increased *apelin* expression in the ISVs in untreated conditions (*Figure 5—figure supplement 2*). We confirmed our results using in situ hybridization (*Figure 5—figure supplement 3*). These observations indicate that *apelin* expression is affected in *tie1* mutants as well as *svep1* morphants, and support the hypothesis of Tie1 and Svep1 acting in the same molecular pathway.

## *tie2* loss of function does not exacerbate the *tie1* mutant phenotype

To investigate a possible contribution of Tie2 to lymphatic Tie signalling as well as possible compensatory mechanisms, we examined *tie1; tie2* double mutants at 2 dpf (*Figure 6A–G*). While *tie1* mutants showed a highly significant reduction in PL numbers (*Figure 6D, G*), we found that an additional loss of one or two functional copies of *tie2* did not further affect PL numbers in *tie1* mutant embryos (*Figure 6E–G*). Additionally, loss of one *tie1* allele in *tie2* mutants did not result in any defects (*Figure 6C, G*). To further exclude contributions of Tie2 at later stages of lymphatic development on TD formation, we quantified the segments of TD across 10 consecutive trunk segments at 5 dpf. In line with our analysis at 2 dpf, heterozygous loss of *tie1* did not reveal any defects in *tie2* mutants (*Figure 6H*). These results therefore do not support a role of *tie2* in zebrafish lymphatic development.

## Genetic interaction between *svep1* and *tie1* during PL migration in the trunk

After having excluded a potential role for Tie2 during lymphangiogenesis, and given the high phenotypic similarity between *tie1* and *svep1* mutants, we wondered whether both genes might act in the same pathway during lymphangiogenesis and would therefore show a genetic interaction. To this end, we quantified PL cell numbers in embryos from *svep1; tie1* double heterozygous parents at 2 dpf. In *svep1; tie1* double heterozygous embryos we could not observe any PL number reduction, reduction of BLECs or reduced facial lymphatics compared to siblings (*Figure 7A, B, H*; *Figure 7—figure supplement 1*), while *tie1* and *svep1* single mutants again showed severe reduction of PL cell numbers (*Figure 7C, D, H*). Importantly, these defects were significantly exacerbated in *svep1^{+/−}*;

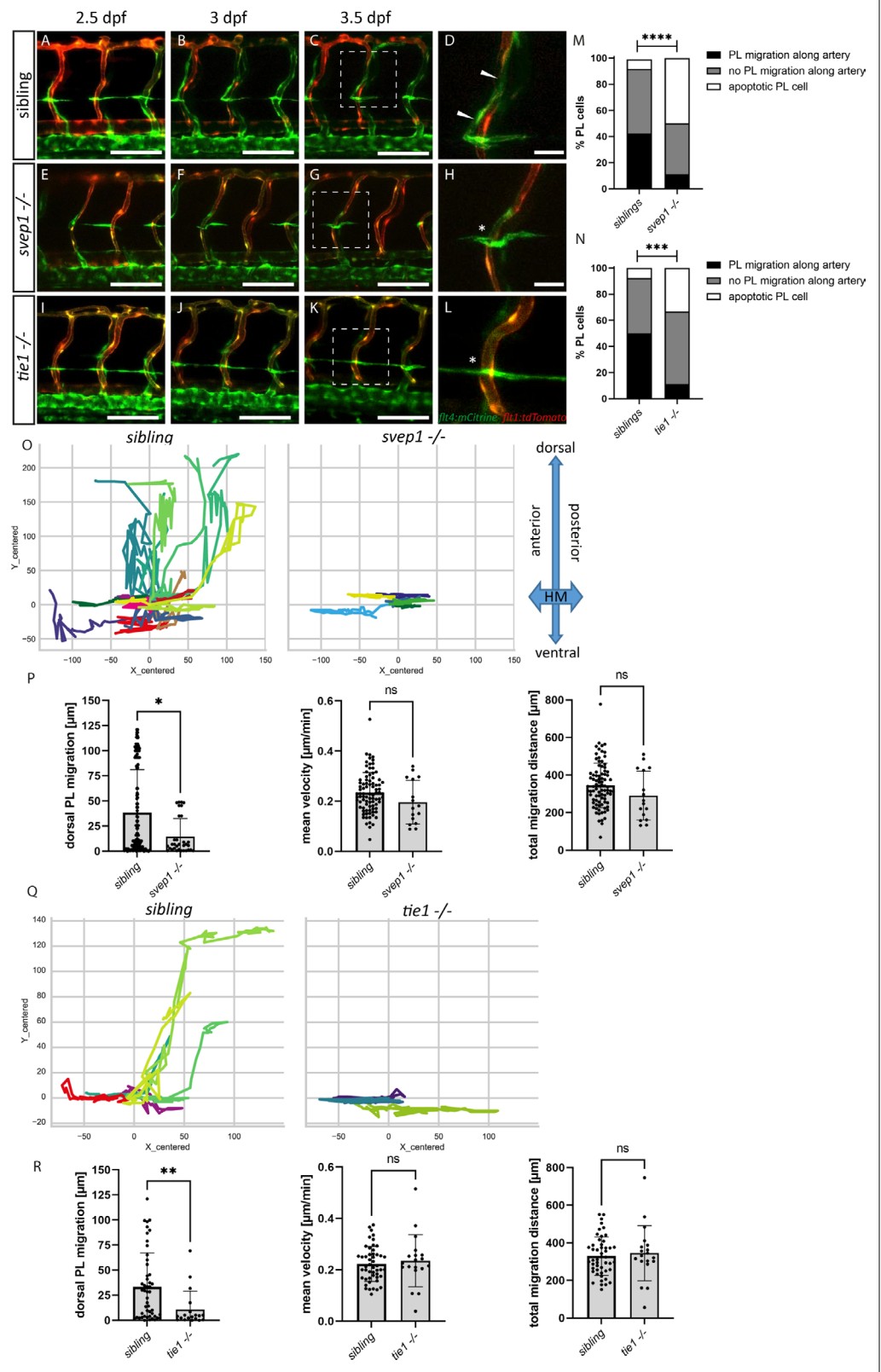

**Figure 4.** PL cell migration along arteries is severely affected in *svep1* and *tie1* mutants. (**A–L**) Still frames from confocal time-lapse imaging of embryos in a *flt4:mCitrine; flt1:tdTomato* transgenic background. (**A–D**) PL migration (indicated by arrowheads) of sibling embryo along aISV from 2.5 to 3.5 dpf. (**E–H**) Failed PL migration (indicated by asterisk) of *svep1* mutants and (**I–L**) *tie1* mutants along artery from 2.5 to 3.5 dpf. (**M, N**) Classification

*Figure 4 continued on next page*

*Figure 4 continued*

of PL migration along arteries. Statistical analysis was performed using Mann–Whitney test comparing the % of PL migration along arteries in each sibling and mutant embryo (*sibling*: n = 96 PLs in 18 embryos; *svep1−/−*: n = 36 PLs in 15 embryos; siblings: n = 52 PLs in 14 embryos; *tie1−/−*: n = 28 PLs in 10 embryos); ****p < 0.0001, ***p = 0.0003. (**O, Q**) Representative cell tracking routes (tracks centred to origin) of single PL cells marked by different colours in siblings (n = 17 PLs in 4 embryos; n = 7 in 2 embryos), *tie1−/−* (n = 5 PLs in 2 embryos) and *svep1−/−* (n = 6 PLs in 3 embryos). (**P, R**) Quantification of dorsal and ventral PL migration (delta Y migration distance), mean velocity and total migration distance in *svep1* and *tie1* mutants compared to sibling embryos excluding apoptotic PLs quantified in (**M, N**) revealed decreased migration in dorsal and ventral direction in *svep1* (*p = 0.0148) as well as *tie1* mutants (**p = 0.0023). ns = not significant; aISV, arterial intersegmental vessel; dpf, days post fertilization; HM, horizontal myoseptum; PL, parachordal lymphangioblast. Scale bar = 100 µm (D, H, L = 25 µm).

The online version of this article includes the following video and figure supplement(s) for figure 4:

**Figure supplement 1.** *svep1* mutants display PL migration defect.

**Figure supplement 2.** *tie1* mutants display PL migration defect.

**Figure 4—video 1.** Confocal time-lapse imaging from 2.5 until 3.5 dpf in the trunk of sibling embryos positive for *flt4.mCitrine* and *flt1:tdTomato*.
https://elifesciences.org/articles/82969/figures#fig4video1

**Figure 4—video 2.** Confocal time-lapse imaging from 2.5 until 3.5 dpf in the trunk of *svep1* mutant embryos positive for *flt4.mCitrine* and *flt1:tdTomato*.
https://elifesciences.org/articles/82969/figures#fig4video2

**Figure 4—video 3.** Confocal time-lapse imaging from 2.5 until 3.5 dpf in the trunk of *tie1* mutant embryos positive for *flt4.mCitrine* and *flt1:tdTomato*.
https://elifesciences.org/articles/82969/figures#fig4video3

*tie1−/−* compared to *svep1+/+; tie1−/−* mutant embryos (*Figure 7D, F, H*). In *svep1−/−, tie+/−* mutant embryos, we observed a tendency of fewer PLs compared to *svep1* single mutants (*Figure 7C, E, H*). However, this effect was not significant. Taken together, this interaction study strengthens the idea that Svep1 converges in the Tie1 pathway.

## Svep1 is a binding ligand of Tie1

To interrogate whether Svep1 and Tie1 bind directly, we performed biochemical analyses using co-immunoprecipitation of Svep1 and Tie1 proteins. We co-transfected zebrafish Svep1 with zebrafish Tie1 or Tie2 constructs and detected Tie1 and Tie2 after immunoprecipitation of Svep1. Tie1 showed robust binding with Svep1 in every experiment (seven out of seven independent experiments; *Figure 8A*), while Tie2 co-precipitated with Svep1 in only two out of four experiments (*Figure 8—figure supplement 1A*). These results demonstrate that zebrafish Tie1 constitutes a binding partner for zebrafish Svep1. We performed the same experiment with human proteins, this time using only the C-terminus of human SVEP1 (aa: 2261–3571). We could observe that TIE1 showed binding to SVEP1, demonstrating that the SVEP1/TIE1 interaction is evolutionary conserved (*Figure 8B*). Additionally, we observed SVEP1 association with TIE2 (*Figure 8—figure supplement 1B*). In a second approach, we used purified SVEP1 protein (C-terminus) and the lysates of TIE1 or TIE2 transfected cells, to confirm our results and to obtain a better impression of the respective binding affinities. After pull-down of SVEP1, we detected binding of SVEP1 with TIE1 (*Figure 8C*) but no significant binding of TIE2 (*Figure 8—figure supplement 1C*). Therefore, we conclude that TIE1 is a ligand for SVEP1.

## Discussion

We had previously demonstrated a role for Svep1 in formation of functional lymphatic vessels in mice and zebrafish. We here extend this phenotypic analysis and show an essential role for zebrafish Svep1 during formation of specific aspects of the facial lymphatic network and of BLECs. Additionally, we uncover a crucial role for Tie1 signalling during lymphangiogenesis and DLAV formation under reduced flow conditions in zebrafish and provide strong in vivo evidence for *svep1* and *tie1* interaction. We also show direct binding of SVEP1 to TIE1 in vitro for the respective human and zebrafish proteins. The results thus establish Svep1 as a factor in Tie1 signalling in zebrafish, both in lymphatic and blood vascular beds. *svep1* mutants display a very specific phenotype in the facial lymphatic

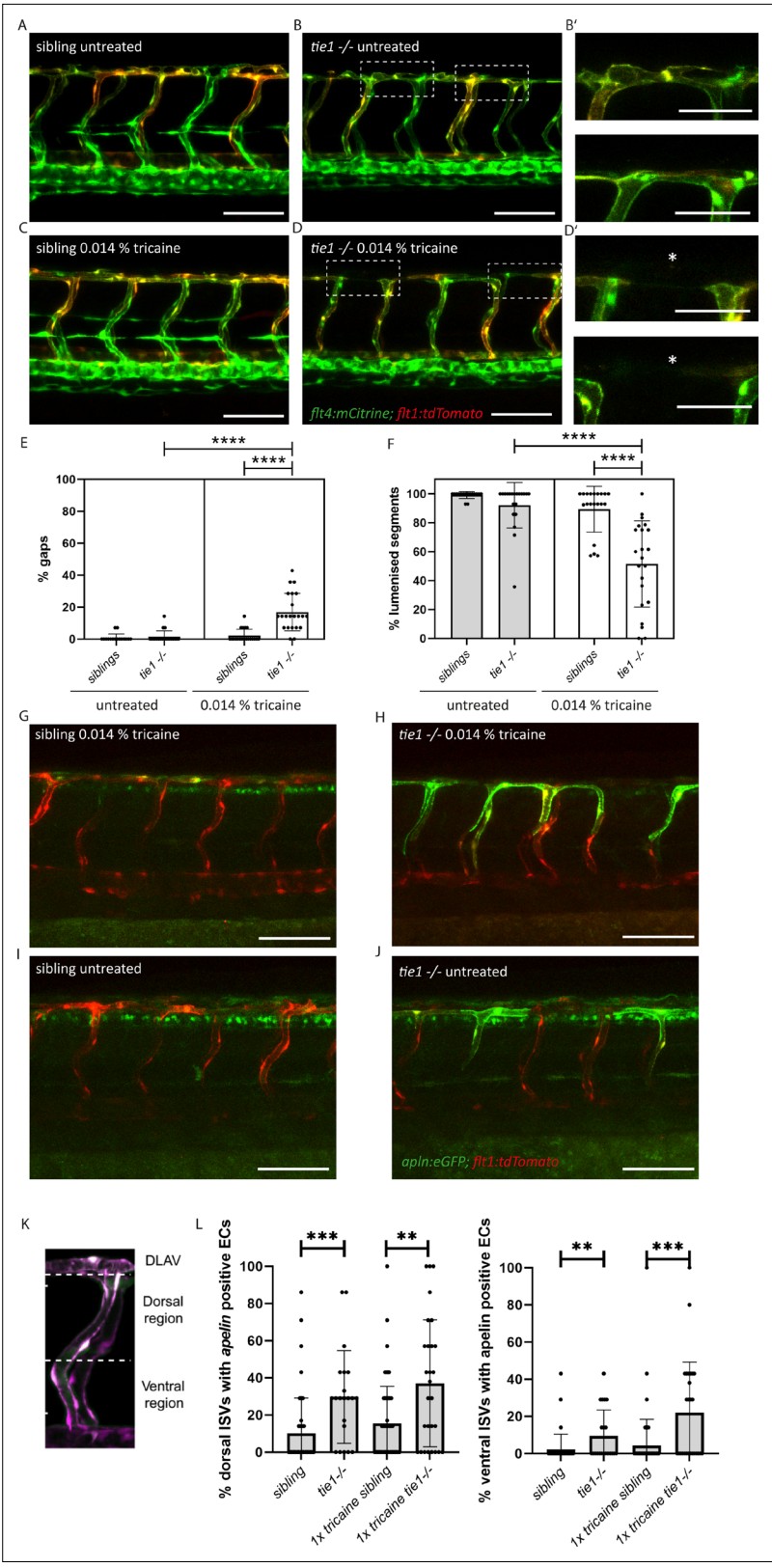

**Figure 5.** Reduced blood flow leads to vascular anastomosis defects in *tie1* mutants, similar to the defects in *svep1* mutants. (**A, B**) Confocal images of sibling and *tie1* mutant embryos at 2 dpf in a *flt4:mCitrine* and *flt1:tdTomato* transgenic background. (**B′**) Magnification and reduced stack of boxed area in (**B**). (**C, D**) Confocal images of *sibling* and *tie1* mutant embryos treated with 0.014% tricaine from 30 until 48 hpf. Asterisks indicate incompletely

*Figure 5 continued on next page*

*Figure 5 continued*

formed DLAV segments. (**D'**) Magnification and reduced stack numbers of boxed area in (**D**). (**E**) Quantification of gaps in the DLAV in sibling and *tie1* mutants that were either untreated or treated with 0.014% tricaine revealed significant increase of gaps in the DLAV in *tie1* mutants. (**F**) Quantification of lumenized trunk segments of the DLAV in siblings and *tie1* mutants, either untreated or treated with 0.014% tricaine (siblings untreated: $n = 16$; *tie1−/−* untreated: $n = 20$; siblings treated with 0.014% tricaine: $n = 20$; *tie1−/−* treated with 0.014% tricaine: $n = 22$), revealed significant decrease of lumenized segment numbers in the DLAV in *tie1* mutants. Mann–Whitney test was applied for statistical analysis. (**G, H**) *apelin:eGFP* and *flt1:tdTomato* expression in 48-hpf-old embryos after tricaine treatment from 30 to 48 hpf and (**I, J**) in untreated conditions. (**K**) Maximum intensity projection of an aISV at 48 hpf, highlighting the ventral and dorsal region used for further quantifications in (**J**) adapted from Figure 5J of *Coxam et al., 2022*. (**L**) Quantification of ISVs with *apelin* expression in dorsal and ventral parts of the ISVs. Dorsal part was counted from DLAV until midline region. Lateral region was counted from midline region onwards in ventral direction. *tie1* mutants showed significant increase of *apelin* positive ECs compared to siblings in untreated (dorsal: ***$p = 0.0001$; ventral: **$p = 0.0028$) and treated with 0.014% tricaine conditions (dorsal: **$p = 0.0033$; ventral: ***$p = 0.0002$) (siblings untreated: $n = 53$; *tie1−/−* untreated: $n = 21$; siblings treated with 0.014% tricaine: $n = 66$; *tie1−/−* treated with 0.014% tricaine: $n = 28$). Mann–Whitney test was applied for statistical analysis. Values are presented as means ± standard deviation (SD). ****$p < 0.0001$. Scale bar = 100 μm. hpf, hours post-fertilization; ISV, intersegmental vessel; DLAV, dorsal longitudinal anastomotic vessel; dpf, days post-fertilization.

The online version of this article includes the following figure supplement(s) for figure 5:

**Figure supplement 1.** *vegfc* and *tie2* mutants do not show defects in DLAV formation upon tricaine treatment.

**Figure supplement 2.** *svep1* morphants show increased *apelin* expression in ISVs.

**Figure supplement 3.** *apelin* expression is reduced in *svep1* and *tie1* mutants.

---

bed, which is distinct from, and complementary to the phenotypes we observed in mutants of Vegfc/Vegfr3 pathway members (*Figure 1*). Previous studies demonstrated that mutations in *vegfc*, *ccbe1*, and *adamts3; adamts14* lead to a complete loss of the facial lymphatic vasculature. These studies did not assess the effects on the recently described FCLV (*Astin et al., 2014*; *Okuda et al., 2012*; *Padberg et al., 2017*; *Wang et al., 2020*). In the present study, we show that the FCLV is still formed in mutants affecting the Vegfc/Vegfr3 signalling cascade, whereas mutations in either *svep1* or *tie1* result in a near-complete loss of this structure. In line with a differential requirement for *svep1/tie1* for the development of the facial lymphatic vessels and the FCLV, we found that *svep1* is expressed in close proximity to the lymphatic sprout arising from the PHS giving rise to the FCLV, while *vegfc* is expressed in cells that appear to be predominantly positioned around the migration route of the FLS arising from the CCV (*Figure 1C*). Based on this highly specific mutant phenotype, we conclude that Svep1 is essential for FCLV formation in a Vegfc-independent manner. Therefore, we here show for the first time that besides the previously postulated functional and morphological differences between FCLV and the facial lymphatics (*Shin et al., 2019*), there is also a difference in the pathways controlling the formation of both structures. Until recently, it was traditionally considered, that lymphatic vessels always (1) have a venous origin and (2) need Vegfc signalling to develop. In the last decade, it was shown that lymphatic vessels can also have non-venous origins in mice (*Martinez-Corral et al., 2015*) and also in the facial lymphatics of zebrafish (*Eng et al., 2019*). However, Vegfc signalling seemed to be always required for lymphatic vessel development. Interestingly, inactivation of *Angpt1* and *Angpt2* or *Tie2* completely abolishes Schlemm's canal development and leads to glaucoma formation in mice, while the Schlemm's canal is still present and only reduced in mice lacking *Vegfc* and *Vegfd* or *Vegfr3*, indicating that in some lymphatic structures VEGFC is not strictly required (*Bernier-Latmani and Petrova, 2017*; *Thomson and Quaggin, 2018*). Here, we make the significant finding that a specific progenitor population of zebrafish facial lymphatic network, forming the FCLV, develops in a Vegfc-independent, but Svep1/Tie1-dependent mechanism. Since the Schlemm's canal is a hybrid vessel (*Kizhatil et al., 2014*) and the FCLV seems to be also morphological and functional different from other lymphatic vessels (*Shin et al., 2019*), these two vessels not only share mechanistical but also functional differences to other lymphatic structures.

While the majority of mutants identified in forward genetic lymphatic screens affect known or novel members of the Vegfc signalling pathway with highly similar phenotypes, the *svep1* mutants stand out due to phenotypic differences compared to mutants affecting the *Vegfc/Vegfr3* pathway. This, however, raises the question how Svep1 exerts its effects during lymphangiogenesis. In the current

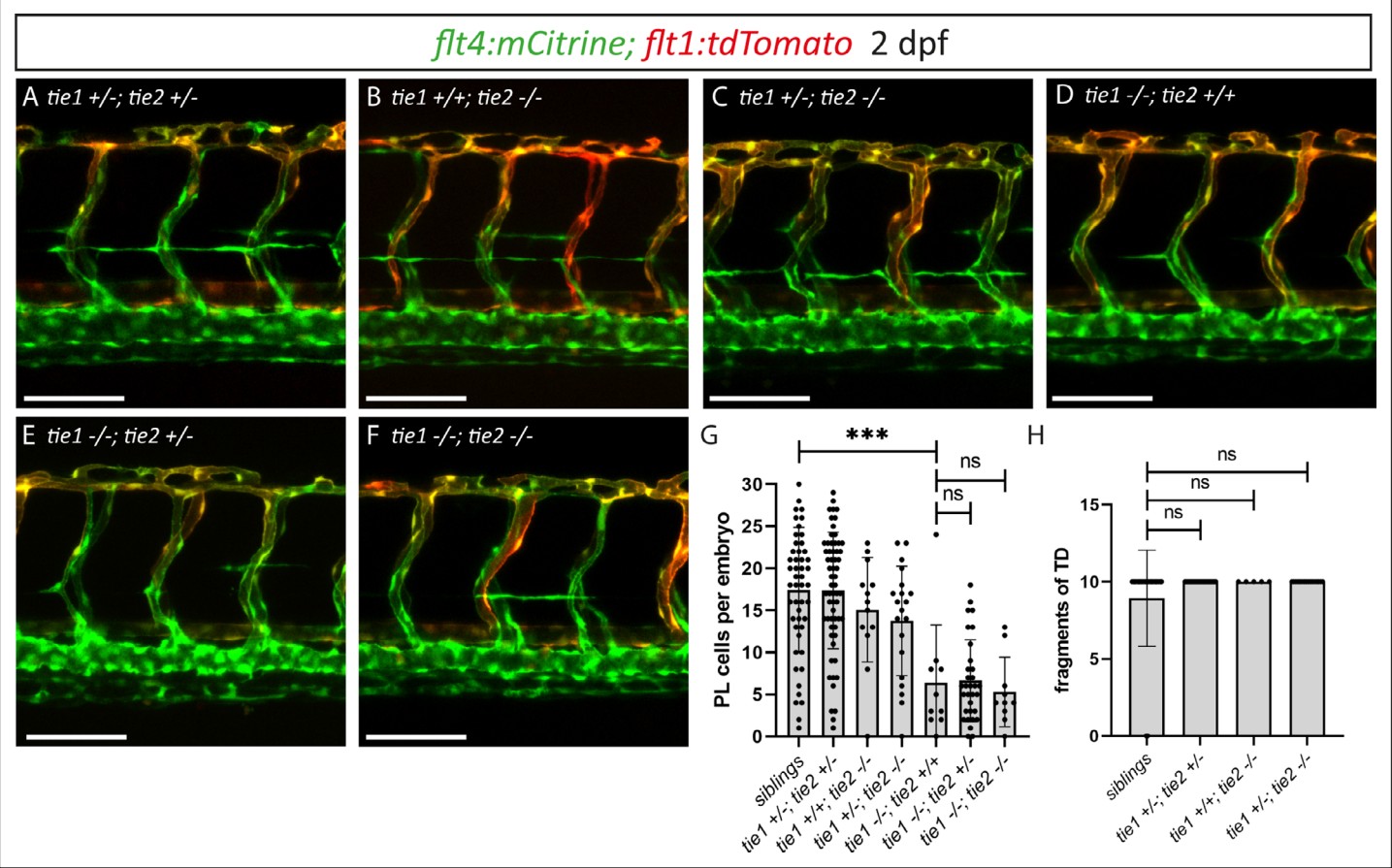

**Figure 6.** *tie1; tie2* double mutants show no exacerbation of the *tie1* mutant defects. (**A–F**) Confocal images of blood and lymphatic vasculature in the trunk of 2 dpf old embryos derived from *tie1; tie2* double heterozygous parents, showing no genetic interaction between *tie1* and *tie2*. (**G**) Quantification of PLs at 2 dpf and of thoracic duct fragments at 5 dpf (siblings: *n* = 50; *tie1*+/−; *tie2*+/−: *n* = 62; *tie1*+/+; *tie2*−/−: *n* = 13; *tie1*+/−; *tie2*−/−: *n* = 20; *tie1*−/−; *tie2*+/+: *n* = 10; *tie1*−/−; *tie2*+/−: *n* = 32; *tie1*−/−; *tie2*−/−: *n* = 10). (**H**) TD fragments were counted over the anterior-most 10 somites (siblings: *n* = 47; *tie1*+/−; *tie2*+/−: *n* = 34; *tie1*+/+; *tie2*−/−: *n* = 5; *tie1*+/−; *tie2*−/−: *n* = 16). Mann–Whitney test was applied for statistical analysis. ***p = 0.0002, ns = not significant. Scale bar = 100 µm. dpf, days post-fertilization; PL, parachordal lymphangioblast; TD, thoracic duct.

study, we focused on a potential connection to the Tie signalling pathway, as murine ANG1 and ANG2 had been shown to bind Svep1 in vitro (*Morooka et al., 2017*). In mice, conditional knockout of *Svep1* or *Tie2* leads to high intraocular pressure and altered Schlemm's canal morphology (*Li et al., 2020*; *Thomson et al., 2014*). Additionally, *Tie2* and *Tie1* expression levels are downregulated in *Svep1* mutant mice (*Morooka et al., 2017*).

While *Tie2* knockout mice display severe cardiovascular defects and die at E9.5 (*Dumont et al., 1994*; *Sato et al., 1995*), *tie2* mutant zebrafish show unaltered vascular structures including unaffected trunk lymphatics (*Gjini et al., 2011*; *Jiang et al., 2020*). We here extend this notion to lymphatic structures in the head of the embryo: as is the case for PL cell numbers, neither the formation of facial lymphatics nor of BLECs depend on Tie2 activity. Teleost *tie2* has actually been lost in the Acanthomorphata lineage, comprising 60% of contemporary teleost species (*Jiang et al., 2020*), suggesting either the loss of critical Tie2 function in most teleosts, or the adoption of essential functions for mammalian TEK function within the last 450 million years (*dos Reis et al., 2015*). This complicates functional comparison between mammalian and teleost Ang/Tie signalling. *Tie1* mutant mice do not show any vascular defects until E13.5 and die from haemorrhages between E13.5 and P0, but display swellings at E12.5 caused by lymphatic malformations that precede the haemorrhaging (*D'Amico et al., 2010*; *Puri et al., 1995*; *Sato et al., 1995*). Additionally, postnatal *Tie1* deletion causes impaired lymphatic capillary network development (*Korhonen et al., 2022*). We here show that *tie1* mutant zebrafish embryos display severe lymphatic defects in the head and trunk vasculature, in addition to the previously reported cardiac and blood vascular phenotypes including impaired

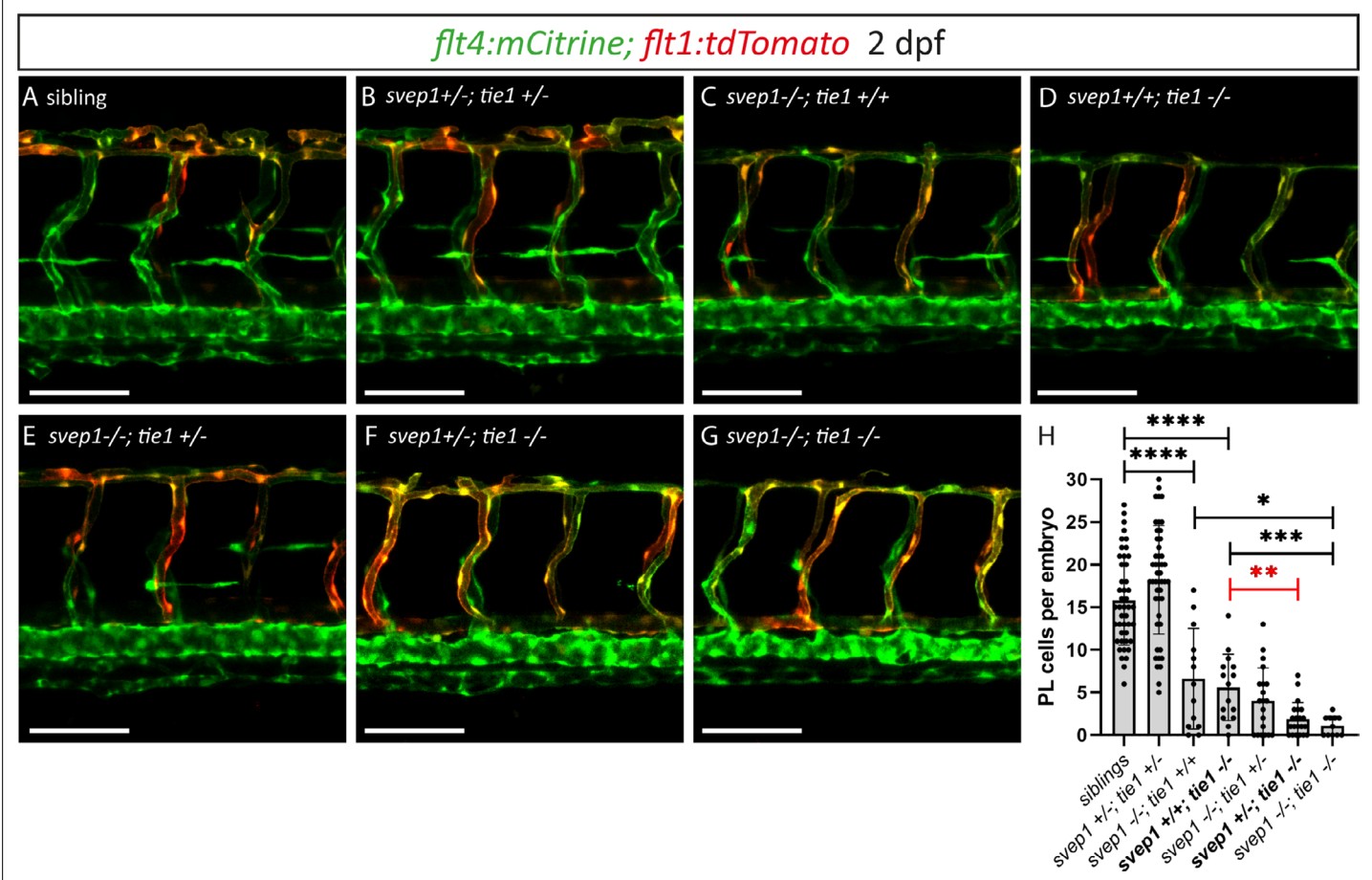

**Figure 7.** Heterozygous loss of *svep1* exacerbates the PL phenotype in *tie1* mutants, indicating genetic interaction between *svep1* and *tie1*. (**A–G**) Confocal images of blood and lymphatic vasculature in the trunk of 2-dpf-old embryos derived from *svep1; tie1* double heterozygous fish, showing severely reduced PL numbers in *svep1; tie1* double mutants and significant decrease of PL cell numbers in *svep1+/−; tie1−/−* compared to *svep1+/+; tie1−/−* (**p = 0.0012). (**H**) Quantification of PL cell numbers at 2 dpf using Mann–Whitney test (siblings: *n = 45*; *svep1+/−; tie1+/−*: *n = 45*; *svep1−/−; tie1+/+*: *n = 13*; *svep1+/+; tie1−/−*: *n = 15*; *svep1−/−; tie1+/−*: *n = 20*; *svep1+/−; tie1−/−*: *n = 21*; *svep1−/−; tie1−/−*: *n = 11*). Scale bar = 100 μm. Values are presented as means ± standard deviation (SD), ****p < 0.0001, ***p = 0.007, *p = 0.0163, ns = not significant. dpf, days post-fertilization; PL, parachordal lymphangioblast.

The online version of this article includes the following figure supplement(s) for figure 7:

**Figure supplement 1.** *svep1; tie1* double heterozygouse embryos show normal BLECs and facial lymphatics.

**Figure supplement 2.** *tie1* expression is not altered in *svep1* mutants.

brain angiogenesis, reduced CCV width, and impaired caudal vein plexus formation (*Carlantoni et al., 2021*). Interestingly, the FCLV, which seems to have a comparable function to collecting lymphatic vessels, is affected in *tie1* mutant zebrafish embryos, while *Tie1;Tie2* double deletion in mice leads to defective postnatal collecting lymphatic vessel development (*Korhonen et al., 2022*). Further studies will be required in both mice and fish to determine to what extent Tie signalling affects LEC specification, proliferation, and survival. However, we here show definitively that Tie signalling is not only required in mice and humans for lymphatic vessel formation, but also in zebrafish.

Remarkably, lymphatic and non-lymphatic defects observed in *tie1* mutant zebrafish embryos are very similar to the defects observed in *svep1* mutants: while reduced PL numbers and TD length is a hallmark feature of many lymphatic mutants, the specific absence of the FCLV is unique, and common to both mutants. Furthermore, formation of BLECs is affected in both mutants, and the specific PL migration phenotype, with PL cells at the HM not migrating dorsally or ventrally, is also observed in both *svep1* and *tie1* mutants (*Figure 4*). In addition, we could show that *tie1* mutants show similar vascular defects in DLAV formation under reduced flow conditions compared to *svep1* mutants, while

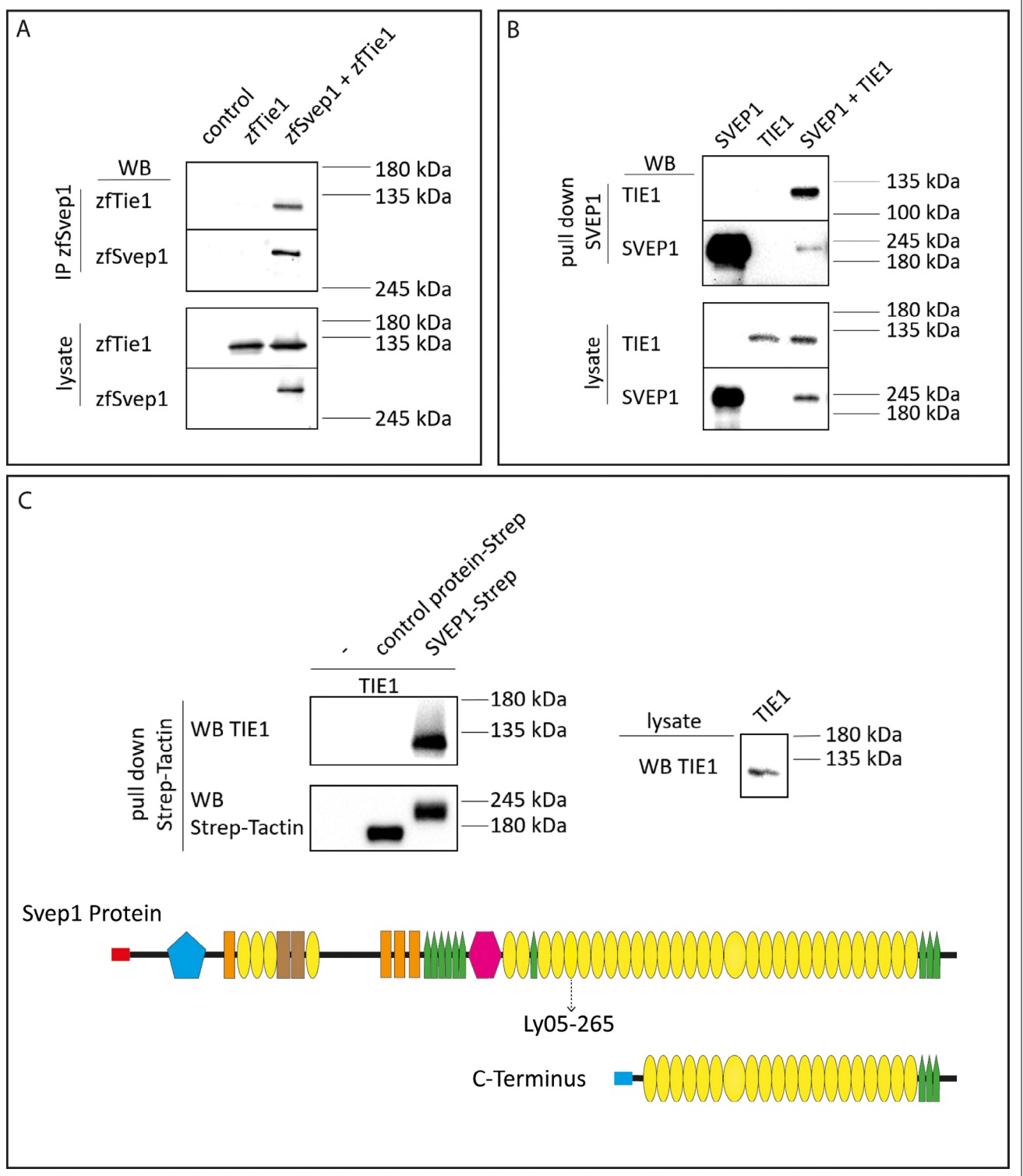

**Figure 8.** SVEP1 binds to TIE1. (**A**) 293T HEK cells were transfected with zebrafish Svep1-HIS (zfSvep1) and zebrafish Tie1-HA (zfTie1). zfSvep1 was immunoprecipitated and associated Tie1 was detected by western blot. (**B**) Co-immunoprecipitation of C-terminal human SVEP1 co-transfected in 293T HEK cells with human TIE1. (**C**) Pull-down of recombinant C-terminal human SVEP1-Strep-tag II protein, which was incubated with TIE1 transfected 293T HEK cell lysates, shows binding of TIE1. Protein structure with all domains indicated and C-terminal part used for pull-down assays (adapted from Figure 2F of *Karpanen et al., 2017*, published under the CC BY-NC 4.0 license, https://creativecommons.org/licenses/by-nc/4.0/). It is not covered by the CC-BY 4.0 license and further reproduction of this panel would need to follow the terms of the CC BY-NC 4.0 license. Ly05-265 indicates position of stop codon in the zebrafish hu6985 allele (*Karpanen et al., 2017*), suggesting that the protein domains C-terminal to the nonsense allele are critical

*Figure 8 continued on next page*

*Figure 8 continued*

for function. Red and blue rectangle: signal peptide; blue pentagon: von Willebrand factor type A domain (vWF); orange rectangle: ephrin-receptor like domain; brown rectangle: Hyalin repeat; yellow ovals: SUSHI repeat; green pentagons: epidermal growth factor (EGF)-like and calcium-binding EGF-like domains; and pink hexagon: pentraxin domain (PTX).

The online version of this article includes the following source data and figure supplement(s) for figure 8:

**Source data 1.** Raw data of western blots.

**Figure supplement 1.** Co-transfected Tie2/TIE2 precipitates with Svep1/SVEP1, but does not bind recombinant human C-terminal SVEP1.

**Figure supplement 1—source data 1.** Raw data of western blots.

we did not observe any defects in *vegfc* and *tie2* mutants. Another hallmark of the *svep1* phenotype is the increase in *apelin* expression in the ISVs, which is again recapitulated in *tie1* mutants. Therefore, we conclude that *svep1/tie1* signalling is not only important for lymphangiogenesis but also for blood vessel development and acts, at least to some extent, in a Vegfc-independent manner. Finally, genetic interaction studies indicate that Svep1 provides essential input into the Tie1 pathway, as losing one copy of *svep1* in *tie1* mutants exacerbates the phenotype significantly when assessing PL cell numbers (*Figure 7*). Of note, elimination of both *tie2* alleles did not alter the *tie1* mutant phenotype (*Figure 6*). Since Young et al. reported Svep1 as a possible genetic modifier of *TIE2* (*Young et al., 2020*), and Morooka et al. showed that *Tie1* as well as *Tie2* expression levels are downregulated in *Svep1* deficient mice (*Morooka et al., 2017*), we assessed *tie1* expression levels in zebrafish *svep1* mutants. However, using in situ hybridization, we did not find any signs of miss-regulation of *tie1* expression in *svep1* mutants (*Figure 7—figure supplement 2*), indicating that at least in zebrafish downregulation of *tie1* is not causative for the observed defects.

There are at least two possibilities how Svep1 could exert its effect on Tie1 function. First, and based on the observation that murine Svep1 can bind Tie receptor ligands (*Morooka et al., 2017*), zebrafish Svep1 could be an essential component of enabling or stabilizing binding of angiopoietins to Tie1. Second, and due to the close proximity of *svep1* expressing cells to LECs, Svep1 could also act as a direct ligand for Tie1. Further studies will clarify whether angiopoietins are a required component of Svep1/Tie1 signalling, and to what extent there are species-specific differences between teleosts and mammals.

In summary, we here show that zebrafish Svep1 as well as human SVEP1 can bind to Tie1/TIE1, indicating SVEP1 as a novel binding partner of TIE1. For human SVEP1 we only used the C-terminus for binding assays, since we reasoned, based on a previously generated *svep1* mutant line (Ly05-265) with a premature stop in CCP domain 9 (*Karpanen et al., 2017*), that the C-terminus is critical for Svep1 function. Further studies, such as ELISA or Biacore assays that allow quantitative assessment of binding affinities for TIE1 and TIE2, will be required to allow more qualified statements on whether and to which extend SVEP1 can also bind TIE2. Taken together, we provide the first in vivo and in vitro evidence that Svep1 interacts with Tie1, and that both genes, at least in certain vascular beds, act in a Vegfc-independent manner. Thus, we here clarify the importance of the respective roles of Tie1 as well as Tie2 in zebrafish, but also underline the significance of Svep1 and Tie1 signalling in different vascular beds. Together with the recent discovery that SVEP1 could act as a modifier of TEK-related PCG disease penetrance, further studies in zebrafish can serve as an in vivo model for clinically relevant aspects of Svep1/Tie signalling.

## Materials and methods

**Key resources table**

*Continued on next page*

| Reagent type (species) or resource | Designation | Source or reference | Identifiers | Additional information |
|---|---|---|---|---|
| Genetic reagent (*D. rerio*) | Tg(flt4:mCitrine)*hu7135* | **van Impel et al., 2014** | ZFIN: hu7135 | |
| Genetic reagent (*D. rerio*) | Tg(flt1*enh*:tdTomato)*hu5333* | **Bussmann and Schulte-Merker, 2011** | ZFIN: hu5333 | |
| Genetic reagent (*D. rerio*) | Tg(lyve1:DsRed2)*nz101* | **Okuda et al., 2012** | ZFIN: nz101 | |
| Genetic reagent (*D. rerio*) | Tg(UAS:RFP)*nkuasrfp1a* | **Asakawa et al., 2008** | ZFIN: nkuasrfp1a | |
| Genetic reagent (*D. rerio*) | Tg(vegfc:Gal4FF)*mu402* | **Wang et al., 2020** | ZFIN: mu402 | |
| Genetic reagent (*D. rerio*) | Tg(svep1:GAL4FF)*hu8885* | **Karpanen et al., 2017** | ZFIN: hu8885 | |
| Genetic reagent (*D. rerio*) | adamts3*hu10891* | **Wang et al., 2020** | ZFIN: hu10891 | |
| Genetic reagent (*D. rerio*) | adamts14*hu11304* | **Wang et al., 2020** | ZFIN: hu11304 | |
| Genetic reagent (*D. rerio*) | vegfc*hu6410* | **Helker et al., 2013; Le Guen et al., 2014** | ZFIN: hu6410 | |
| Genetic reagent (*D. rerio*) | ccbe1*hu10965* | **Kok et al., 2015** | ZFIN: hu10965 | |
| Genetic reagent (*D. rerio*) | svep1*hu6123* | **Karpanen et al., 2017** | ZFIN: hu6123 | |
| Genetic reagent (*D. rerio*) | svep1*hu4767* | **Karpanen et al., 2017** | ZFIN: hu4767 | |
| Genetic reagent (*D. rerio*) | tie1*bns208* | **Carlantoni et al., 2021** | ZFIN: bns208 | |
| Genetic reagent (*D. rerio*) | tie2*hu1667* | **Gjini et al., 2011** | ZFIN: hu1667 | |
| Genetic reagent (*D. rerio*) | Tg*BAC*(apln:EGFP)*bns157* | **Helker et al., 2020** | ZFIN: bns157 | |
| Cell line (*Homo sapiens*) | 293T HEK cells | **Roukens et al., 2015** | | |
| Cell line (*Homo sapiens*) | HEK293 EBNA | Manuel Koch | | |
| Transfected construct (*D. rerio*) | zfTie1-HA in PCS2+ | This paper | | Provided by Naoki Mochizuki |
| Transfected construct (*D. rerio*) | zfTie2-HA in PCS2+ | This paper | | zfTie2 cDNA provided by Naoki Mochizuki |
| Transfected construct (*D. rerio*) | zfSvep1-HIS in PCS2+ | This paper | | |
| Transfected construct (*Homo sapiens*) | TIE1-HA in PCEP4 | This paper | | TIE1 cDNA provided by Hellmut Augustin |
| Transfected construct (*Homo sapiens*) | TIE2-HA in PCEP4 | This paper | | Provided by Manuel Koch |
| Transfected construct (*Homo sapiens*) | SVEP1-Strep II in PCEP4 | This paper | | Provided by Manuel Koch |
| Transfected construct (*Mus musculus*) | mouse nope ectodomain with Fc tag strep | This paper | | Provided by Manuel Koch |
| Antibody | anti-HA (rat monoclonal) | Roche | 11867423001 | 1:10,000 |
| Antibody | anti-rat (donkey polyclonal) | Invitrogen | # A18745 | 1:15,000 |

| Reagent type (species) or resource | Designation | Source or reference | Identifiers | Additional information |
|---|---|---|---|---|
| Antibody | anti-HIS (rabbit polyclonal) | Invitrogen | # PA1-983B | 1:250 |
| Antibody | anti-HIS (mouse monoclonal) | Invitrogen | # MA1-135 | 1:250 |
| Antibody | anti-mouse (goat polyclonal) | dako | P0447 | 1:4000 |
| Antibody | anti-DIG primary antibody (sheep polyclonal) | Roche | 11093274910 | 1:2000 |
| Recombinant DNA reagent | apelin in pGEM-t-easy | Provided by Christian Helker | | For in situ probe generation |
| Peptide, recombinant protein | SVEP1-Strep II Purified recombinant protein | This paper | | Provided by Manuel Koch |
| Chemical compound, drug | DIG RNA Labeling Mix | Roche | 11277073910 | |
| Chemical compound, drug | Lipofectamin 2000 transfection reagent | Thermo Fisher Scientific | 11668030 | |
| Chemical compound, drug | FuGENE HD Transfection Reagen | Promega | E2311 | |
| Chemical compound, drug | T4 Ligase | Thermo Fisher Scientific | EL0012 | |
| Chemical compound, drug | FCS | Merck Chemicals GmbH | F7524 | |
| Chemical compound, drug | Q5 Hot Start High-Fidelity DNA Polymerase | New England Biolabs GmbH | M0493 | |
| Chemical compound, drug | DMEM/F-12, GlutaMAX Supplement | Thermo Fisher Scientific | 10565018 | |
| Chemical compound, drug | Strep-Tactin Superflow high capacity resin | IBA Lifesciences GmbH | 2-1208-002 | |
| Software, algorithm | GraphPad Prism 6 | GraphPad Software, USA | | |
| Software, algorithm | Fiji-ImageJ (version 1.52); Manual tracking plugin; StrackReg plugin | DOI:10.1038/nmeth.2019; Fabrice Cordelières, Institut Curie, Orsay (France); DOI:10.1109/83.650848 | RRID:SCR_002285 | |
| Software, algorithm | Python (version 3.8) | Python.org | RRID:SCR_008394 | Code available at https://github.com/MuensterImagingNetwork/Hussmann_et_al_2022 |
| Other | Strep-Tactin HRP conjugate | Iba-lifesciences | 2-1502-001 | 1:10,000 |

## Zebrafish strains and husbandry

Animal work followed guidelines of the animal ethics committees at the University of Münster, Germany, and fish were maintained following FELASA guidelines (*Aleström et al., 2020*). The following transgenic and mutant lines have been used in this study: *Tg(flt4:mCitrine)$^{hu7135}$* (*van Impel et al., 2014*), *Tg(flt1$^{enh}$:tdTomato)$^{hu5333}$* (*Bussmann and Schulte-Merker, 2011*), *Tg(lyve1:DsRed2)$^{nz101}$* (*Okuda et al., 2012*), *Tg(UAS:RFP)$^{nkuasrfp1a}$* (*Asakawa et al., 2008*), *Tg(vegfc:Gal4FF)$^{mu402}$* (*Wang et al., 2020*), *Tg(svep1:GAL4FF)$^{hu8885}$* (*Karpanen et al., 2017*), *adamts3$^{hu10891}$* (*Wang et al., 2020*), *adamts14$^{hu11304}$* (*Wang et al., 2020*), *vegfc$^{hu6410}$* (*Helker et al., 2013*; *Le Guen et al., 2014*), *ccbe1$^{hu10965}$* (*Kok et al., 2015*), *svep1$^{hu6123}$* (*Karpanen et al., 2017*), *svep1$^{hu4767}$* (*Karpanen et al., 2017*) (only used for *svep1;ccbe1* double knockout, *Figure 1—figure supplement 2*), *tie1$^{bns208}$* (*Carlantoni et al., 2021*), *tie2$^{hu1667}$* (*Gjini et al., 2011*), and *Tg$^{BAC}$(apln:EGFP)$^{bns157}$* (*Helker et al., 2020*).

## Genotyping

For genotyping of *svep1*, *adamts3*, *adamts14*, *vegfc*, and *tie2*, KASPar (Biosearch Technologies) was used, and for *ccbe1* and *tie1* High-Resolution Melt Analysis (*Samarut et al., 2016*; *Supplementary file 1*).

## Live imaging and microscopy

Live imaging was carried out on 2, 3, and 5 dpf embryos. Before 24 hpf, 1-phenyl-2-thiourea (75 mM, Sigma, #P7629) was added to inhibit melanogenesis (*Karlsson et al., 2001*). For imaging, embryos were anesthetized with 42 mg/l tricaine (Sigma, #A5040) and embedded in 0.8% low melting agarose (Thermo Fischer, #16520100) dissolved in embryo medium. Embryo medium containing tricaine was layered on top of the agarose once solidified for overnight imaging. Additionally, embryos were kept at 28°C during overnight imaging. Embryos were imaged with an inverted Leica SP8 microscope using a ×20/×0.75 dry objective or a ×40/1.1 water immersion objective detection and employing Leica LAS X 3.5.7.23225 software. Scoring of PLs or TD fragments was performed using a Leica M165 FC and an X-Cite 200DC (Lumen Dynamics) fluorescent light source. Confocal stacks were processed using Fiji-ImageJ version 1.52 g. Brightfield images were taken using an Olympus SZX16 microscope and a LEICA DFC450 C camera. Images and figures were assembled using Adobe Illustrator. All data were processed using raw images with brightness, colour, and contrast adjusted for printing.

## Cell tracking

To quantify the migration distance and mean velocity of the PLs from 2.5 to 3.5 dpf, the leading edge of each PL was manually tracked using 'Manual Tracking'-Plugin (Fabrice Cordelières, Institut Curie, Orsay (France)) in Fiji-ImageJ (version 1.52 g source, *Schindelin et al., 2012*). For image stabilization 'StackReg' using rigid body (*Thevenaz et al., 1998*) was applied to the maximum intensity projections of the time-lapse movies prior to manual tracking. Mean track velocity and total migration distance (sum of all leading edge displacements) were calculated using a custom Python script (version 3.8). To plot the migration route, track start coordinates were centred to the origin and individual cell tracks were represented using a line plot (Python). Y PL migration was defined as the absolute value of the distance in Y direction (dorsal and ventral) from track origin to the last tracking point (ΔY). Scripts used for data analysis are available at GitHub. Data were analyzed using GraphPad for plotting and statistical analysis.

## Tricaine treatment

Where applicable, embryos were treated with 0.014% tricaine (Sigma, #A5040) from 30 to 48 hpf to slow down heart rate and blood flow during DLAV formation as previously described (*Coxam et al., 2022*).

## In situ hybridization

Antisense RNA probes of *tie1* were generated from amplified cDNA. Primers for cDNA generation are listed in *Supplementary file 1*. Antisense RNA probes of *apelin* were generated from cDNA kindly provided by Christian Helker (*Helker et al., 2015*). Since the reverse primer contained a T3 overhang, we proceeded with in vitro transcription using T3 RNA polymerase and digoxigenin (DIG)-labelled UTP (2 hr at 37°C). Fixation of 24 hpf embryos from a *svep1* heterozygous incross was performed with 4% paraformaldehyd (PFA) overnight at 4°C. In situ hybridization was performed according to previous published protocols using 100 ng of each of the respective probes (*Schulte-Merker, 2002*). Staining procedure was monitored regularly over time to ensure proper detection of differences in staining intensities between embryos.

## Cloning and expression of human SVEP1, TIE1, and TIE2

The C-terminal part of SVEP1 was amplified from human cDNA using Q5 polymerase and cloned into the sleeping beauty transposon system (*Kowarz et al., 2015*; NM_153366.4 aa: 2261–3571; N-terminal BM-40 signal peptide followed by a Twin-Strep-tag). After verification of the plasmid by sequencing, the expression construct was co-transfected with the transposase plasmid (10:1) into HEK293 EBNA cells (tested negative for mycoplasma) using FuGENE HD transfection reagent (Promega GmbH, Madison, USA) in DMEM/F12 supplemented with 6% fetal bovine serum. After high puromycin selection (3 µg/ml; Sigma), cells were expanded in triple flasks and protein production induced with doxycycline (0.5 µg/ml, Sigma). Supernatants of confluent cells were harvested every 3 days, filtered and recombinant proteins purified via Strep-TactinXT (IBA Lifescience, Göttingen, Germany) resin. SVEP1 was eluted with biotin-containing buffer (IBA Lifescience, Göttingen, Germany), dialyzed against TBS and stored at 4°C or −80°C. The human sequences of TIE1 (NP_005415.1 aa: 21–1138) and

TIE2 (NP_000450.3 aa: 23–1124) were cloned into the PCEP episomal expression system (transient) including an HA-tag sequence at the C-terminal part in the reverse primers. For the PCR amplification, TIE2 was amplified from human cDNAs, and TIE1 from a plasmid kindly provided by Hellmut Augustin.

## In vitro binding assay

For co-immunoprecipitation we first transfected 293T HEK cells with zebrafish Svep1-HIS, zebrafish Tie1-HA, zebrafish Tie2-HA, human SVEP1 (C-Terminus)-StrepII, human TIE1-HA or human TIE2-HA as well as the indicated combinations using Lipofectamin 2000 reagent. After 48 hr the cell lysates were collected using Ripa buffer (50 mM Tris pH 7.5, 1% NP-40, 0.1% sodium dodecyl sulfate, 0.5% Na-deoxycholate, 150 mM NaCl). For co-immunoprecipitation of zebrafish Svep1-HIS, the cell lysates were incubated for 1 hr with 30 µl G-Sepharose beads (17061801, GE Healthcare) and 3 µg of anti-HIS antibody (# PA1-983B, Invitrogen). For pull-down of human SVEP1-StrepII, we used Strep-TactinXT 4Flow high capacity resin (2-5030-025, iba-lifesciences). Afterwards, the beads were washed five times with Ripa buffer and boiled for 5 min at 95°C in sample buffer.

In an independent approach, the C-terminus of human StrepII tagged SVEP1 protein (amino acids 2261–3571) was generated and purified. This protein and a StrepII tagged control protein (3 µg) were incubated with 50 µl Strep-TactinXT 4Flow high capacity resin in 500 µl binding buffer (50 mM Tris–HCl at pH 7.5, 100 mM NaCl, 0.02% Triton X-100) for 30 min and added to the cell lysate of TIE1-HA and TIE2-HA transfected cells. After 2 hr incubation, the beads were washed with Ripa buffer and processed like the co-transfection samples. All samples were subjected to western blot analysis using anti-HA high affinity antibody (11867423001, Roche) to detect co-precipitated TIE1 and TIE2. The respective secondary antibody was HRP conjugated and detected using Lumi-Light Western Blotting Substrate (12015200001, Roche) and ChemiDoc MP Imaging System (Biorad). For western blot analysis of zfSvep1-HIS we used anti-His mouse antibody (# MA1-135, Invitrogen) and for SVEP1 we used Strep-Tactin HRP conjugate (2-1502-001, iba-lifesciences).

| antibody | dilution | provider |
|---|---|---|
| anti-HA rat | 1:10000 | 11867423001, Roche |
| anti-rat donkey | 1:15000 | # A18745, Invitrogen |
| anti-HIS rabbit | 1:250 | # PA1-983B, Invitrogen |
| anti-HIS mouse | 1:250 | # MA1-135, Invitrogen |
| anti-mouse goat | 1:4000 | P0447, dako |
| Strep-Tactin HRP conjugate | 1:10000 | 2-1502-001, iba-lifesciences |

## Statistics and reproducibility

Data sets were tested for normality (Shapiro–Wilk) and equal variance p-values of data sets with normal distribution were determined by Welch's *t*-test or Student's *t*-test. In case data values did not show normal distribution, a Mann–Whitney test was performed instead. All statistical tests were performed using GraphPad Prism 8 or Microsoft Excel. All experiments were carried out at least two times. Only tricaine treatment of *vegfc* mutants (*Figure 5—figure supplement 1*) was carried out once.

## Acknowledgements

This work was supported by CRC 1348 (DFG, project B08 to M.H. and S.S.-M.) and by the CiM-IMPRS graduate school. The original human TIE1 plasmid was kindly provided by Hellmut Augustin.

## Additional information

### Competing interests

Didier YR Stainier: Senior editor, *eLife*. The other authors declare that no competing interests exist.

## Funding

| Funder | Grant reference number | Author |
|---|---|---|
| Deutsche Forschungsgemeinschaft | CRC 1348 project B08 | Melina Hußmann Stefan Schulte-Merker |

The funders had no role in study design, data collection, and interpretation, or the decision to submit the work for publication.

## Author contributions

Melina Hußmann, Conceptualization, Data curation, Formal analysis, Validation, Investigation, Visualization, Methodology, Writing - original draft, Project administration, M.H. performed all experiments, with support from D.S. in the biochemistry work; Dörte Schulte, Data curation, Supervision, Methodology, Writing - review and editing; Sarah Weischer, Resources, Data curation, Software, Visualization, Writing - review and editing; Claudia Carlantoni, Hiroyuki Nakajima, Naoki Mochizuki, Didier YR Stainier, Resources, Writing - review and editing; Thomas Zobel, Resources, Software, Writing - review and editing; Manuel Koch, Resources, Investigation, Methodology, Writing - review and editing, M.K. generated final forms of all human constructs, and recombinant SVEP1 protein; Stefan Schulte-Merker, Conceptualization, Resources, Supervision, Funding acquisition, Investigation, Writing - original draft, Project administration, Writing - review and editing

## Author ORCIDs

Melina Hußmann ⓘ http://orcid.org/0000-0003-4798-7503
Sarah Weischer ⓘ http://orcid.org/0000-0001-7292-8308
Claudia Carlantoni ⓘ http://orcid.org/0000-0003-3716-8539
Naoki Mochizuki ⓘ http://orcid.org/0000-0002-3938-9602
Didier YR Stainier ⓘ http://orcid.org/0000-0002-0382-0026
Stefan Schulte-Merker ⓘ http://orcid.org/0000-0003-3617-8807

## Decision letter and Author response

Decision letter https://doi.org/10.7554/eLife.82969.sa1
Author response https://doi.org/10.7554/eLife.82969.sa2

# Additional files

## Supplementary files

• Supplementary file 1. Primer list for genotyping, cloning, and in situ probe generation.

• MDAR checklist

## Data availability

Scripts used for data analysis available at GitHub at https://github.com/MuensterImagingNetwork/Hussmann_et_al_2022 (copy archived at *Münster Imaging Network, 2023*). Source Data files have been provided for Figures 8 and Figure 8 figure supplement 1.

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
