## [Editor Report]

This study presents strong and compelling evidence that the extra-cellular matrix protein SVEP-1 interacts with the TIE1 receptor to promote aspects of lymphangiogenesis that are independent of canonical VEGF-C signaling. Using zebrafish models to show genetic interactions and cells to provide evidence of biochemical interaction, the study shows a functional requirement for these genes/proteins in specific aspects of lymphangiogenesis. These novel findings will be of interest to developmental and cell biologists and to those studying lymphatic disease as it potentially provides novel therapeutic targets.

---

## [Decision Letter]

**Decision letter after peer review:**

Thank you for submitting your article "svep1 and tie1 genetically interact and affect aspects of facial lymphatic development in a Vegfc-independent manner" for consideration by *eLife*. Your article has been reviewed by 3 peer reviewers, one of whom is a member of our Board of Reviewing Editors, and the evaluation has been overseen by Richard White as the Senior Editor. The following individual involved in the review of your submission has agreed to reveal their identity: Heinz-Georg Belting (Reviewer #2).

Essential revisions:

1. The loss of the FCLV with relevant genetic manipulations should be verified using a second vascular marker/reporter, to overcome the caveat that perhaps flt4 expression is selectively affected.

2. The apelin link should be strengthened via in situ hybridization to complement the reporter readout and the claim that apelin is a downstream target modified unless direct evidence can be provided.

3. The work would benefit by determining if double heterozygosity for tie1 and svep1 affects the relevant lymphangiogenic structures/processes.

4. While not required, we encourage the authors to consider providing biochemical data from in vitro experiments to complement the genetic interactions between Tie1/svep1 described in the fish model. This would substantially strengthen the work and allow more discussion of the mechanism. Absent such data the lack of molecular/biochemical mechanism to accompany the genetic readouts should be explicitly stated.

*Reviewer #3 (Recommendations for the authors):*

Recommendations for experiments to strengthen the claims:

1. Loss of FCLV in svep1 and tie1 mutants is shown using one marker (flt4:mCitrine). Can the authors demonstrate loss of FCLV using another marker (or tracer injection), to exclude that reduced flt4 expression leads to a failure to visualize this structure in the mutants? What is the functional consequence of the loss of this specific vessel?

2. The conclusion that Svep1 constitutes a component of the Tie1 pathway is based on similarities between the mutant phenotypes and genetic interaction studies but biochemical evidence needed to establish this claim is lacking. Does Svep1 interact with Tie1 directly to induce phosphorylation and downstream signaling? Alternatively, as the authors propose, does Svep1 regulate Angiopoietin availability and/or activity, thereby inducing Tie signaling? in vitro experiments would help to answer some of these questions.

3. Genetic interaction between svep1 and tie1 is shown in the regulation of PL numbers and migration. Do double heterozygous svep1;tie1 mutants also show other vascular defects observed in svep1 and tie1 mutants, including in FCLV formation?

4. The authors show that Apelin expressing ECs were increased in the ISVs of tie1 mutant, similar to what was shown for svep1 morphants treated with tricaine. Based on this, they conclude that Apelin is a downstream target of Tie1 and Svep1, however, no molecular evidence is provided. Can they provide such evidence, to exclude that increased Apelin expression is secondary to vascular defects in these animals?

Other comments:

1. The authors mention that tie1 embryos develop edema at 4 dpf and were therefore analyzed at an earlier stage. Since svep1 mutants were analyzed at 5 dpf (Figure 1), I assume that they do not show edema. Please comment on this difference.

2. Supplementary Figure 4: please specify what the different graphs represent (Individual embryos?).

---

## [Author Response]

Essential revisions:1. The loss of the FCLV with relevant genetic manipulations should be verified using a second vascular marker/reporter, to overcome the caveat that perhaps flt4 expression is selectively affected.

In Figure 3 figure supplement 1, we now show the loss of the FCLV in a lyve1:DsRed transgenic line to confirm our observations in the flt4:mCitrine transgenic background.

2. The apelin link should be strengthened via in situ hybridization to complement the reporter readout and the claim that apelin is a downstream target modified unless direct evidence can be provided.

We did carry out in situ hybridization, as suggested by the reviewer. Upregulation of *apelin* mRNA levels was confirmed in *svep1* and *tie1* mutants, thus fully supporting our initial data (Figure 5 figure supplement 3). We apologize, however, if we have given the impression that we consider *apelin* necessarily as a downstream target of Svep1/Tie1. This was not our intent, and we have changed the text accordingly.

3. The work would benefit by determining if double heterozygosity for tie1 and svep1 affects the relevant lymphangiogenic structures/processes.

In Figure 7 figure supplement 1 we show that double heterozygosity for *tie1* and *svep1* is likely not affecting the facial lymphatic and BLECs development. We have put the focus in the manuscript on PL cell quantification, as this is a quantifiable population of single cells. For the FCLV, such quantification (cell number, or FCLV volume) is much more difficult to quantify in the transgenic backgrounds that we have available.

4. While not required, we encourage the authors to consider providing biochemical data from in vitro experiments to complement the genetic interactions between Tie1/svep1 described in the fish model. This would substantially strengthen the work and allow more discussion of the mechanism. Absent such data the lack of molecular/biochemical mechanism to accompany the genetic readouts should be explicitly stated.

We now include biochemical data, and have assessed possible binding of the respective Svep1/SVEP1 and Tie1/TIE1 protein version from both zebrafish and humans. The data are now presented in Figure 8, and fully support interaction of the C-terminal half of zebrafish and human Svep1/SVEP1 with the respective Tie1/TIE1 receptors.

Experiments were carried out via transfection experiments and Co-IP in HEK293T cells. In addition, we also challenged human TIE1 with a purified SVEP1 purified protein fragment, confirming transfection results. We trust that these new data indeed considerably strengthen the work, as indicated by the reviewers.

Reviewer #3 (Recommendations for the authors):Recommendations for experiments to strengthen the claims:1. Loss of FCLV in svep1 and tie1 mutants is shown using one marker (flt4:mCitrine). Can the authors demonstrate loss of FCLV using another marker (or tracer injection), to exclude that reduced flt4 expression leads to a failure to visualize this structure in the mutants? What is the functional consequence of the loss of this specific vessel?

We analyzed the facial lymphatics in a lyve1a:DsRed transgenic background in Figure 3 figure supplement 1 and observed the same defects within the FCLV. As far as functional consequences are concerned, the embryos start to develop edema at 5dpf to varying degrees, hence functional assays have proven difficult to interpret.

2. The conclusion that Svep1 constitutes a component of the Tie1 pathway is based on similarities between the mutant phenotypes and genetic interaction studies but biochemical evidence needed to establish this claim is lacking. Does Svep1 interact with Tie1 directly to induce phosphorylation and downstream signaling? Alternatively, as the authors propose, does Svep1 regulate Angiopoietin availability and/or activity, thereby inducing Tie signaling? in vitro experiments would help to answer some of these questions.

See main comments above and new Figure 8.

3. Genetic interaction between svep1 and tie1 is shown in the regulation of PL numbers and migration. Do double heterozygous svep1;tie1 mutants also show other vascular defects observed in svep1 and tie1 mutants, including in FCLV formation?

In Figure 7 figure supplement 1 we show that double heterozygosity for *tie1* and *svep1* is not affecting the facial lymphatic and BLECs development. Of note, double heterozygosity also does not affect PL cell numbers.

4. The authors show that Apelin expressing ECs were increased in the ISVs of tie1 mutant, similar to what was shown for svep1 morphants treated with tricaine. Based on this, they conclude that Apelin is a downstream target of Tie1 and Svep1, however, no molecular evidence is provided. Can they provide such evidence, to exclude that increased Apelin expression is secondary to vascular defects in these animals?

Thank you for bringing this up. We did not mean to suggest this, since we do not have the appropriate evidence. We merely use increase of *apelin* expression as a phenotypic feature that *tie1* and *svep1* mutants share. We have changed the text accordingly, removing any wording that would indicate epistatic relationships between the genes.

Other comments:1. The authors mention that tie1 embryos develop edema at 4 dpf and were therefore analyzed at an earlier stage. Since svep1 mutants were analyzed at 5 dpf (Figure 1), I assume that they do not show edema. Please comment on this difference.

*Svep1* mutants do show edema but not as severe as in *tie1* mutants and not in all embryos with equal severity. It is the reason, however, why we also carried out the genetic interaction analysis in PL cells prior to day 3, in order to stay clear of any possible secondary effects.

2. Supplementary Figure 4: please specify what the different graphs represent (Individual embryos?).

The graphs represent the migration routes of single PL cells in each experiment (corrected in figure legend).